# Threats of global warming to the world's freshwater fishes

Valerio Barbarossa [1,2,3 ✉], Joyce Bosmans[1], Niko Wanders [4], Henry King[5], Marc F. P. Bierkens [4,6],
Mark A. J. Huijbregts [1] & Aafke M. Schipper[1,2]

Climate change poses a significant threat to global biodiversity, but freshwater fishes have been largely ignored in climate change assessments. Here, we assess threats of future flow and water temperature extremes to ~11,500 riverine fish species. In a 3.2 °C warmer world (no further emission cuts after current governments' pledges for 2030), 36% of the species have over half of their present-day geographic range exposed to climatic extremes beyond current levels. Threats are largest in tropical and sub-arid regions and increases in maximum water temperature are more threatening than changes in flow extremes. In comparison, 9% of the species are projected to have more than half of their present-day geographic range threatened in a 2 °C warmer world, which further reduces to 4% of the species if warming is limited to 1.5 °C. Our results highlight the need to intensify (inter)national commitments to limit global warming if freshwater biodiversity is to be safeguarded.

[1] Department of Environmental Science, Institute for Water and Wetland Research, Radboud University, Nijmegen, The Netherlands. [2] PBL Netherlands Environmental Assessment Agency, The Hague, The Netherlands. [3] Institute of Environmental Sciences (CML), Leiden University, Leiden, The Netherlands. [4] Department of Physical Geography, Utrecht University, Utrecht, The Netherlands. [5] Unilever R&D, Safety and Environmental Assurance Centre, Sharnbrook, UK. [6] Deltares, Utrecht, The Netherlands. ✉email: v.barbarossa@cml.leidenuniv.nl

Freshwater habitats are disproportionally biodiverse. While they cover only 0.8% of the Earth's surface, they host ~15,000 fish species, corresponding to approximately half of the global known fish diversity[1,2]. Freshwater habitats are also disproportionally threatened by human activities and environmental change, which have resulted in substantial declines in freshwater biodiversity over the past decades[1,3]. Amid human pressures on freshwater ecosystems (including water abstraction, diversion, damming, and pollution), anthropogenic climate change is expected to become increasingly important in the future[4,5]. Rising air temperatures and changing precipitation patterns modify water temperature and flow regimes worldwide, thus affecting two key habitat factors for freshwater species[6]. Being ectotherms, fish are directly influenced by water temperature, while the hydrologic regime determines the structure and dynamics of the freshwater habitat[7,8]. In addition, the insular nature of many freshwater habitats may hamper compensatory movements to cooler locations, especially for fully aquatic organisms like fish[1]. Recent continental and global studies have underscored the high vulnerability of freshwater fish species to climate change[8–11]. Yet, potential impacts of climate change on freshwater fishes have not yet been comprehensively assessed, in sharp contrast with the many studies assessing potential climate change impacts on species in terrestrial systems[12–15].

Here, we assess future climate threats to 11,425 riverine fish species by quantifying their exposure to flow and water temperature extremes under different global warming scenarios. We focus on extremes rather than hydrothermal niche characteristics in general, because extremes are more decisive for local extinctions and potential geographic range contractions[16,17]. Following the latest IPCC report[18], we include scenarios that limit global mean temperature increases to 1.5 and 2.0 °C. For comparison purposes, we include two additional scenarios: a "current pledges" scenario set at 3.2 °C warming and a "no-policy" scenario (no mitigation) set at 4.5 °C warming (all temperatures relative to pre-industrial)[12,19]. The 3.2 °C warming scenario represents the maximum warming predicted to occur by the end of the century (with 66% probability) if all current greenhouse gas emissions reductions targets (unconditional Nationally Determined Contributions) for 2030 are met and no further cuts are performed. We calculate the present and future weekly flow and water temperature values corresponding with each warming level at a spatial resolution of 5 arcminutes (~10 km) using a global hydrological model coupled to a dynamic water temperature model[20,21]. We force the hydrological model with meteorological input from five Global Climate Models (GCMs) combined with four Representative Concentration Pathway (RCP) representing future greenhouse gas emissions. To assess the threat imposed by future climate extremes, we first retrieve flow and water temperature extremes corresponding with the current climate. Across the geographic range of each species, we quantify the maximum and minimum weekly water temperature and flow, as well as the number of zero flow weeks (see "Methods"). We then define the magnitude of threat for each fish species in a given global warming scenario as the proportion of the geographic range of the species where projected extremes in water flow and temperature will exceed those defined based on the current climate within its ranges. We do this for two dispersal assumptions: 'no dispersal' assuming that each fish species is restricted to its current geographical range, and "maximal dispersal" assuming that each fish species can move beyond its current range within a surrounding region delineated by the intersection of watersheds (hard physical boundaries) and freshwater ecoregions (i.e., regions with similar evolutionary history and other ecological factors relevant to freshwater fishes[22]). Finally, we use phylogenetic regression relating the species' threat levels (i.e., the proportion of the range exposed to future climate extremes beyond current levels within the range) for each warming level and dispersal assumption to a suite of relevant species characteristics.

We find clear differences in the magnitude of threat between the different warming scenarios. In a 3.2 °C warmer world, 36% of the species have over half of their present-day geographic range exposed to climatic extremes beyond current levels (no dispersal assumption). This number reduces to 9% of the species in a 2 °C warmer world and to 4% of the species if warming is limited to 1.5 °C. We conclude that for protecting freshwater biodiversity, commitments to limit global warming need to be strengthened.

## Results

**Global patterns of exposure to projected climate extremes**. The scenario without climate-change mitigation policy (+4.5 °C) and without dispersal resulted in at least half of the geographic range threatened by projected climate extremes for 63% (±7%) of the freshwater fish species. Assuming maximal dispersal for the same warming level, the proportion of species with over half of their geographic range threatened decreased to 24% (±13%). The values in brackets represent the standard deviation of the GCM–RCP combinations ensemble for that warming level and dispersal assumption (Supplementary Fig. 1). The proportion of species with more than half of their range threatened was projected to decrease to 8–36% (±3–11%), 1–9% (±1–4%), and 1–4% (±0–2%) for warming levels of 3.2 °C, 2 °C and 1.5 °C, respectively, with the larger values for the no dispersal assumption (Fig. 1).

We found hotspots of future climate threat in tropical, sub-arid and Mediterranean regions (Fig. 2). At low warming levels, hotspots are restricted to small areas within tropical South America, North-East Mexico, southern US, southern Europe, Southern Sahara, central Africa (large lakes), Middle-East, India–Pakistan, South-East Asia, and western Australia. At higher warming levels, hotspots are considerably larger, particularly in South America, southern Europe, India–Pakistan, and Australia. At higher latitudes, threats become prominent only at higher warming levels (3.2, 4.5 °C). Overall, threats are largest in tropical watersheds such as the Amazon, Parana, Tocantis, Niger, Senegal, Zambezi, and Chao-Phraya (Fig. 3; see Supplementary Fig. 4 for a

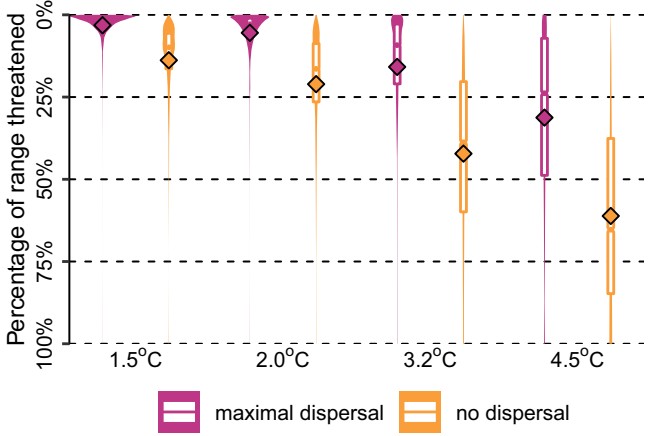

**Fig. 1 Proportion of geographic range threatened at different global warming levels.** The violin plots show the proportion of geographic range threatened by future climate extremes for 11,425 freshwater fish species, different warming levels and two dispersal assumptions. For each species and warming level, the mean across the different scenarios (GCM–RCP combinations) is calculated. Within each violin, the white boxes show the interquartile range as well as the median, while diamonds represent the mean. Source data are provided as a Supplementary Data 1.

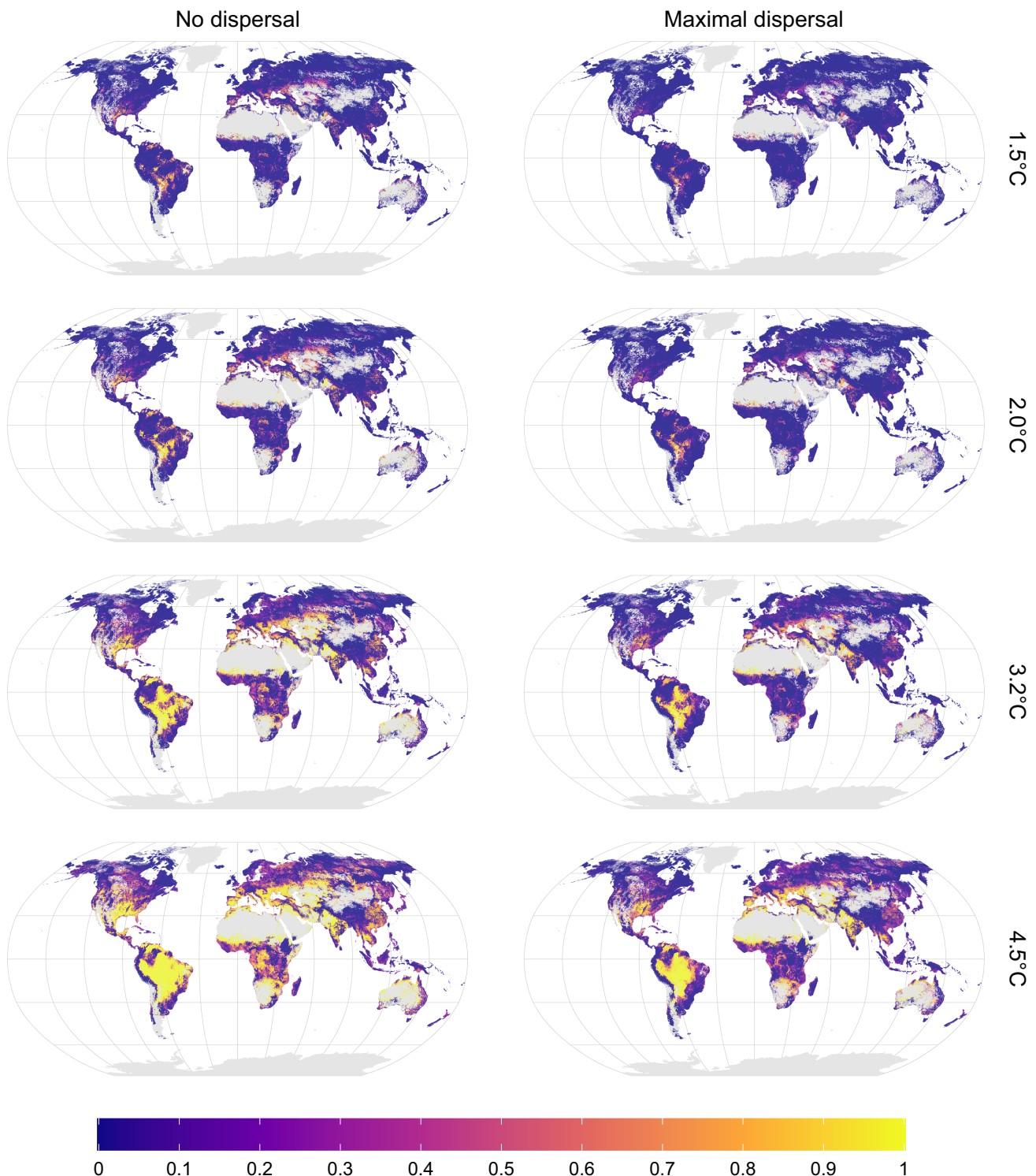

**Fig. 2 Spatial patterns of climate change threat.** Potentially affected fraction (PAF) of freshwater fish species due to exposure to water flow and temperature extremes beyond current levels, for different global warming levels and two dispersal assumptions. Patterns are based on the median PAF across the GCM–RCP combinations at a five arcminutes resolution (~10 km). Gray denotes no data areas (no species occurring or no data available). Source data are provided as Supplementary Data 2 and 3.

more exhaustive overview). Watersheds in non-tropical areas characterized by relatively high threat levels are the Don and the Danube in Europe, and several watersheds in Australia (Fig. 3). Under the maximal dispersal assumption, locations of threatened areas are similar to those under the no dispersal assumption but with lower threat levels than in the no dispersal assumption (Figs. 2 and 3).

**Flow versus water temperature extremes.** Our findings indicate that freshwater fish species are primarily threatened by climate change-induced increases in maximum water temperature, whereas amplified extreme flow conditions are considerably less important (Fig. 4; Supplementary Fig. 2 and Supplementary Tables 3, 4). Projected reductions in minimum water temperature pose virtually no threat (Supplementary Fig. 3 and Supplementary

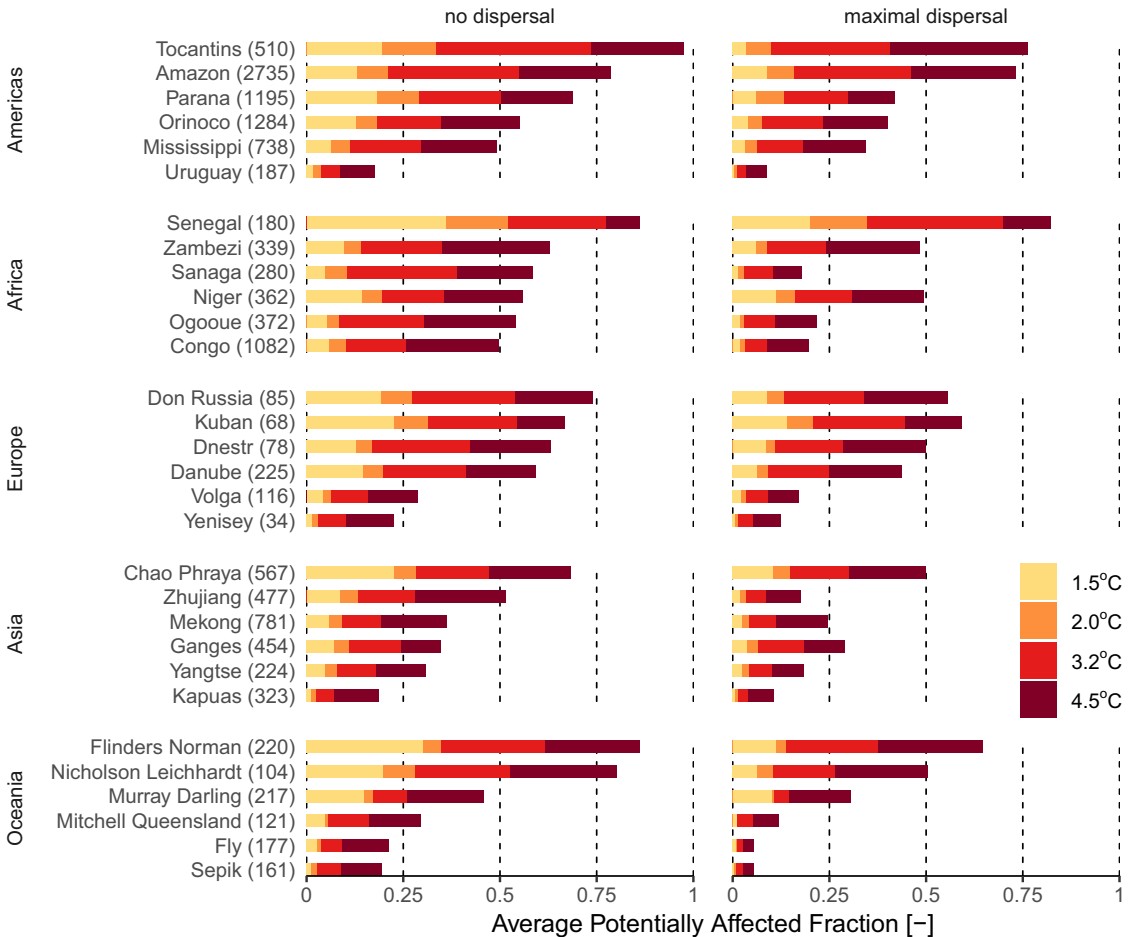

**Fig. 3 Climate change threats in large watersheds.** Potentially affected fraction of species averaged across 5 arcminutes grid cells within 30 large watersheds, for four warming levels and two dispersal assumptions. For each continent defined according to the World Bank Development Indicators (www.worldbank.org), we selected the six watersheds with the largest numbers of species and covering at least 50% of the known species richness (according to Tedesco et al. [2]). A more exhaustive overview of 200 watersheds is available in Supplementary Fig. 4. Numbers in brackets represent the number of species within the watershed. Source data are provided as Supplementary Data 4.

Tables 3, 4). Therefore, the spatial patterns of future climate threat mostly resemble the patterns of threats due to increased maximum water temperature (Fig. 4). Areas affected by changes in low flow are mainly observed in upstream reaches in South America, parts of the central US, around the Mediterranean Sea and Middle East (Fig. 4 and Supplementary Fig. 3). This reflects that climate change is projected to result in more severe low flow conditions mainly in drought-prone regions, while water temperature rises almost everywhere (Supplementary Fig. 5-V). In addition, changes in low flow conditions might be more relevant for smaller upstream streams not captured within the ~10 km grid-cell resolution of the global hydrological model employed in this study[20]. In contrast, areas affected by changes in high flow are confined to a few downstream segments of the main stems of large rivers (Supplementary Fig. 3 and raster layers provided as Supplementary Data 2 and 3). Our results further show only limited overlap of threats imposed by amplified flow and water temperature extremes, reflecting the dissimilar spatial distribution of both threats (Fig. 4 and Supplementary Tables 3, 4).

**Relationships between climate change threats and species traits.** According to our phylogenetic regression models ($n =$ 9,779 species), the magnitude of threat imposed by future climate extremes is mainly related to species' habitat type and current geographic range size, followed by IUCN threat status and body length (Fig. 5). Threats are much lower for species that live across the freshwater and marine realms (note that our projections concern the freshwater environment only, thus ignoring potential climate threats within the ocean; Supplementary Table 5). In line with this, relatively small threat levels are found for orders mostly comprising diadromous species, such as Mugiliformes (mullets), Osmeriformes (smelts), Syngnathiformes (e.g., pipefish), Tetraodontiformes (e.g., pufferfish) and Pleuronectiformes (flatfish) (Supplementary Fig. 6). Further, species with a smaller geographic range and body size are more likely to be threatened by climate change (negative regression coefficients; Supplementary Table 5). We also noticed lower threat levels for species currently belonging to a low IUCN threat category (e.g., "near threatened" or "least concern"; Supplementary Table 5). We found similarly low threat levels for species that are "data deficient" within the IUCN Red List (43% of the 9,779 species analyzed). Future climate threats were only weakly related to climate zone, commercial importance category and trophic group (Fig. 5). The results of the traits analysis were largely consistent between the two dispersal scenarios (Fig. 5). The results were less consistent across the warming levels, whereby the importance of geographic range size dropped considerably at higher warming levels under the no dispersal assumption, while habitat type became more important (Fig. 5).

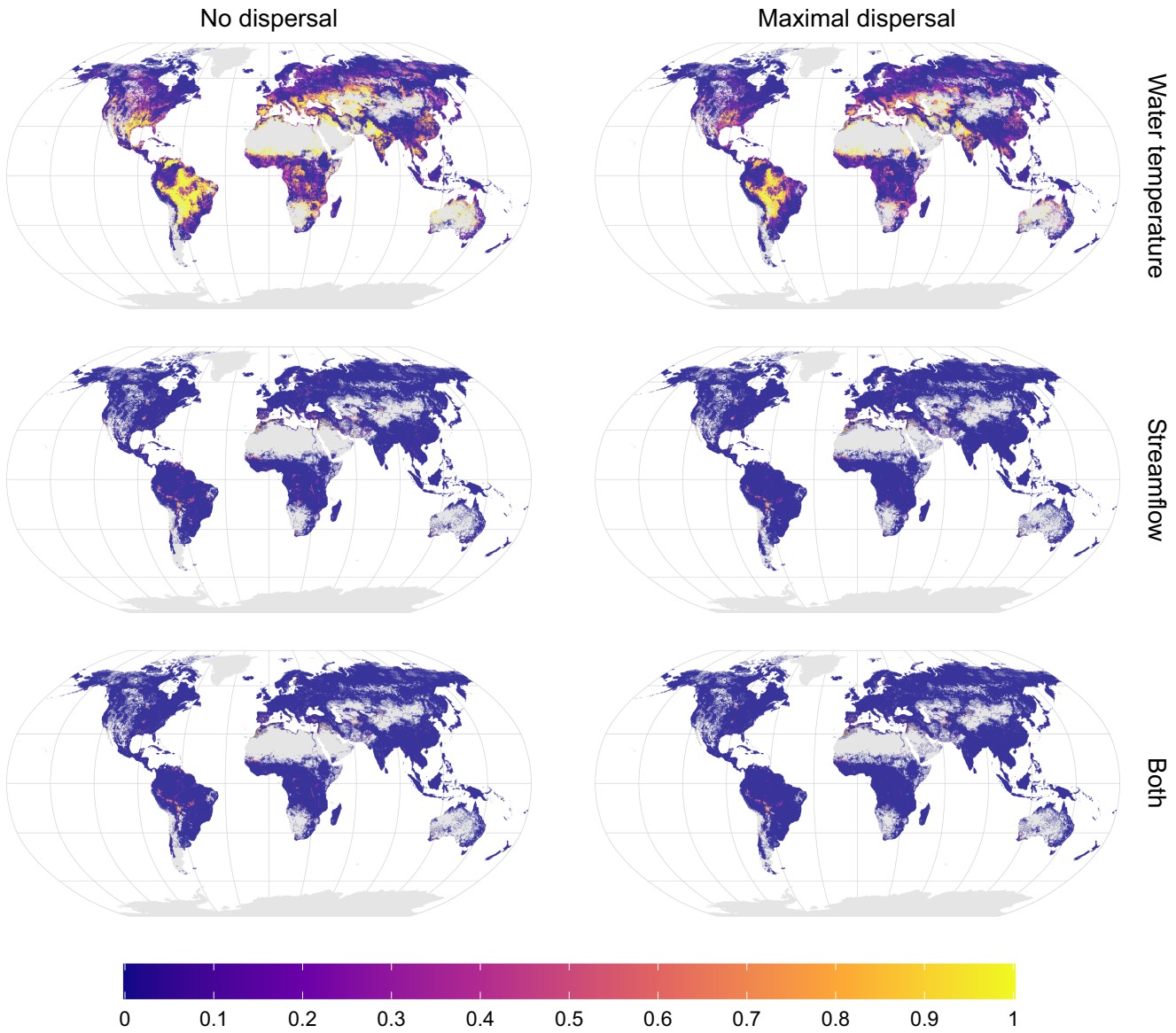

**Fig. 4 Climate change threats due to water temperature versus flow.** Potentially affected fraction (PAF) of species due to changes in water temperature (top), flow conditions (center), or both (i.e., fraction of species threatened by water temperature and flow extremes simultaneously; bottom) for the 3.2 °C warming scenario. The maps represent the median proportion of species affected over the GCM–RCP combinations available for the 3.2 °C warming scenario. Results for the other warming levels are available in Supplementary Fig. 2. A further breakdown of PAF by single variable for the 3.2 °C warming scenario is available in Supplementary Fig. 3. Gray denotes no data areas (no species occurring or no data available). Source data are provided as Supplementary Data 2 and 3.

## Discussion

This study represents the first comprehensive assessment of the threat of potential future climate extremes to freshwater fish species, covering both flow and water temperature, the entire globe and about 90% of the known freshwater fish species. We found that in a "current pledges" scenario (3.2 °C warming), over one third of the freshwater fish species is projected to have more than half of their geographic range threatened by future climate extremes beyond current levels. Threats are considerably reduced under the assumption of maximal dispersal, yet this might represent an overly optimistic estimate given current and future barriers to dispersal[23]. Our results suggest that increases in maximum water temperature constitute a larger threat to freshwater fishes than changes in minimum water temperature or high and low flow conditions. This is because water temperatures vary less within species ranges and are projected to rise almost everywhere, while flow conditions are more

spatially variable hence projected future flow is less likely to exceed present-day extremes within the species' ranges. In line with previous studies, we found that climate change will result in reduced flows mainly in drought-prone regions[21,24]. In addition, depletion of low flows might be most important at low stream orders, which are not well captured by the 5 arcminutes resolution of the hydrological model PCR-GLOBWB employed in this study[20]. While global estimates of hydrological variables are available at higher spatial resolutions[25], 5 arcminutes is the highest resolution currently achievable for future projections of both flow and water temperature[21]. Hence, the spatial resolution of our analysis might result in an underestimation of the impacts of climate change on species living in smaller upstream reaches. We further note that we did not explicitly consider changes in flow or water temperature seasonality, which might disfavor species whose life histories are adapted to specific flow or temperature regimes (e.g., specific seasonal flow

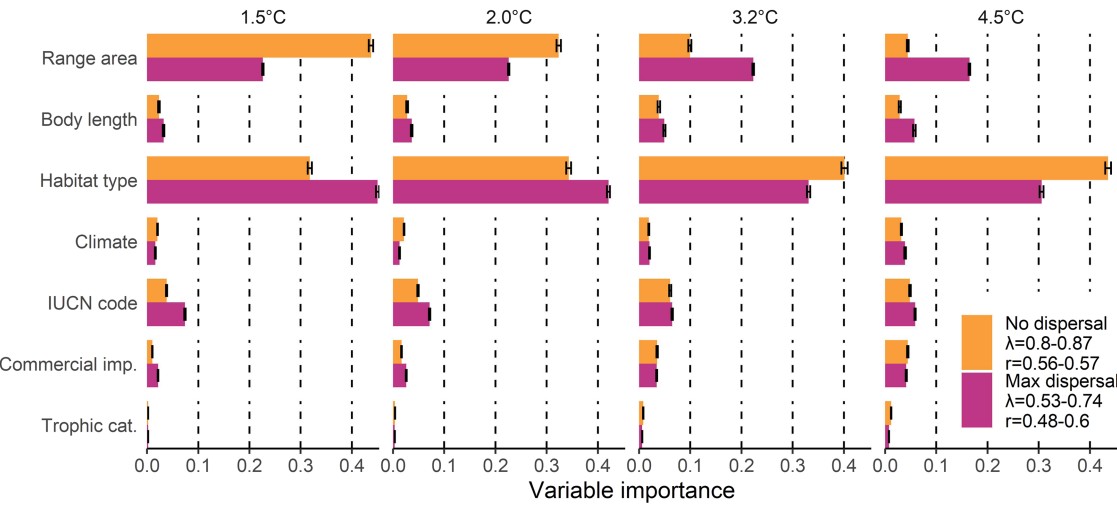

**Fig. 5 Importance of various species properties in explaining the magnitude of future climate threat.** Variables' importance is quantified based on the proportion of the geographic range exposed to climate extremes beyond current levels for two dispersal assumptions. Bars report the mean variable importance and confidence intervals represent the standard deviation across the 100 replicates of the stochastic phylogenetic trees. The legend includes the ranges across the different warming levels in Pagel's $\lambda$ and Pearson's $r$ between the predicted and observed percentage of range threatened. Coefficients of the underlying phylogenetic regression models are presented in Supplementary Table 5. Source data are provided as Supplementary Data 5.

regimes)[5,26]. However, flow and water temperature seasonality were clearly correlated to the minimum flow and water temperature extremes within the species' ranges (Pearson's $r$ of 0.6–0.9), indicating that the effects of changes in the seasonality of flow or temperature were at least partially covered by our predictions.

Exposure to climate extremes beyond present-day values does not necessarily imply local extinction. If species' current distributions are confined by factors other than flow or water temperature (e.g., biogeographic dispersal barriers or anthropogenic pressures), species might be able to withstand larger temperature and flow extremes than inferred based on their current geographic range[27,28]. The same holds if species are able to adapt to new water temperature and flow conditions[16] or if fishes have the possibility to hide from extremes in micro-climatic refugia, for example due to water stratification or small-scale thermal heterogeneity[29], which are not included in our hydrological model[21]. On the other hand, species' range maps are relatively coarse representations of species occurrence, hence some species might be more affected than indicated by our results (i.e., if present-day climate extremes within their geographic range already preclude local occurrence). Indeed, a tentative comparison of our species-specific maximum weekly water temperature limits with critical thermal maxima reported from laboratory tests suggest both under- and overestimations by our geographic range-based thermal limits, while showing an overall reasonable agreement (mean percent difference = 9%; Pearson's $r = 0.62$; Supplementary Fig. 7). Further work is required to better understand deviations between our empirical thresholds and the lab-based maxima, which may stem not only from uncertainties in our modeling approach (e.g., in the range maps or the water temperature model) but also from heterogeneity in experimental conditions[30]. Additionally, we recognize that a given increase in global mean temperature may lead to locally different exposure levels depending on the GCM–RCP combination, as each is characterized by specific distributions of changes in water temperature and flow[31]. We notice a greater variability across GCMs than RCPs when looking at species-specific proportions of range affected (Supplementary Fig. 1), similar to previous findings for hydro-climatic variables[32]. However, variability across the GCMs did not affect the species-specific thresholds, which were consistent across the models (Supplementary Fig. 8).

Although our future climate threat assessment is associated with uncertainties, our comparative analysis across the different warming levels and targets clearly showed a sharp increase in potential impacts with increasing global warming. Species already listed as "endangered" or "critically endangered" on the IUCN Red List of threatened species might be particularly affected by future warming, as these species were characterized by the highest future climate threat levels. Our findings also show that threats imposed by amplified climate extremes are expected to be particularly high in tropical watersheds, in accordance with previous studies suggesting large climate-change induced freshwater habitat degradation in the tropics[33–35]. Tropical species are indeed expected to be highly affected by climate change[36], and our results confirm this for freshwater fishes. Many tropical watersheds host low-income food-deficit countries where local communities are highly dependent on fishery as a primary food source. Indeed, up to 50% of household incomes in countries along the Mekong, Zambezi, and Brazilian Amazon depend on fishing[37]. Hence, increased exposure of freshwater fish species to climate extremes, potentially resulting in local extinctions[17], may have important socio-economic repercussions in these regions[38]. Our findings indicate that limiting global warming to 2 °C will reduce the proportion of freshwater fish species with more than half of their range threatened by 74–81% (the range refers to the two dispersal scenarios) compared to current pledges of governments (3.2 °C). Restricting the global mean temperature rise to 1.5 °C will lower this proportion by an additional 11–14% (or 53-58% compared to 2 °C). While we acknowledge that the ecological realism of our model projections can be improved, these first comparative estimates highlight the need to intensify (inter)national commitments to limit global warming if potentially severe disruption of freshwater biodiversity is to be prevented.

## Methods
**Species occurrence data**. We compiled species' geographic ranges from a combination of datasets. We employed the IUCN Red List of Threatened Species database, which provides geographic range polygons for 8,564 freshwater fish species (~56% of freshwater fish species[39]), compiled from literature and expert knowledge[40]. We complemented these ranges with data from Barbarossa et al[23], who compiled geographic range polygons for 6,213 freshwater fish species not yet represented in the IUCN dataset, and the Amazonfish dataset[41], which provides range maps for 2,406 species occurring in the Amazon basin. We harmonized the species names

based on Fishbase (www.fishbase.org)[42] and merged the ranges (i.e., union of polygons) from the different datasets to obtain one geographic range per species. We then resampled the range polygons of each species to the 5 arcminutes (~10 km) hydrography of the global hydrological model (see below), with a given species marked as occurring in a cell if ≥ 50% of the cell area overlapped with the species' polygon. In total, we obtained geographic ranges for 12,934 freshwater fish species, covering ~90% of the known freshwater fish species[43]. We excluded 1,160 exclusively lentic species because our hydrological model is less adequate for lakes than for rivers, i.e., it does not account for water temperature stratification (see section "Phylogenetic regression on species traits" for an explanation of how habitat information was extracted). Out of the 11,774 (partially or entirely) lotic fish species, we excluded 349 species (~3%) because their occurrence range was smaller than ~1,000 km² (i.e., ten grid cells), which we considered too small relative to the spatial resolution of the hydrological model (see below). Hence, the analysis was based on 11,425 species in total (Supplementary Figs. 9, 10; a raster layer providing the number of species at each five arcminutes grid cell is available as Supplementary Data 6).

**Hydrological data**. We employed the Global Hydrological Model (GHM) PCR-GLOBWB[20] with a full dynamical two-way coupling to the Dynamical Water temperate model (DynWAT)[21] at 5 arcminutes spatial resolution (~10 km at the Equator), to retrieve weekly streamflow and water temperature worldwide[20,21]. PCR-GLOBWB simulates the vertical water balance between two soil layers and a groundwater layer, with up to four land cover types considered per grid cell. Surface runoff, interflow, and groundwater discharge are routed along the river network using the kinematic wave approximation of the Saint–Venant Equations[21] and includes floodplain inundation. Apart from the larger lakes, PCR-GLOBWB includes over 6,000 man-made reservoirs[44] as well as the effects of water use for irrigation, livestock, domestic, and industrial sectors. PCR-GLOBWB computes river discharge, river and lake water levels, surface water levels and runoff fluxes (surface runoff, interflow and groundwater discharge). These fluxes are dynamically coupled to DynWAT along with the meteorological forcing, such as air temperature and radiation from the GCMs to compute water temperature. DynWAT thus includes temperature advection, radiation and sensible heating but also ice formation and breakup, thermal mixing and stratification in large water bodies, effects of water abstraction and reservoir operations. We selected this model combination because it allows a full representation of the hydrological cycle (considering also anthropogenic stressors, e.g., water use), it fully integrates water temperature and calculates the hydrological variables on a high-resolution hydrography. The choice of one hydrological model over an ensemble was motivated by the fact that very few GHMs or Land Surface Models calculate water temperature at the spatial resolution desired for this study[20,21]. The PCR-GLOBWB model setup was similar to Wanders et al.[21], with the exception that flow and water temperature were aggregated at the weekly scale to capture the fish species' tolerance levels to extreme events[45].

**Species-specific thresholds for extreme flow and water temperature**. To assess climate change threats to freshwater fishes, we focused on climate extremes rather than hydrothermal niche characteristics in general, because extremes are more decisive for local extinctions and potential geographic range contractions[16,17]. We quantified climate extremes using long-term average maximum and minimum water temperature ($T_{max}$, $T_{min}$), maximum and minimum flow ($Q_{max}$, $Q_{min}$), and the number of zero flow weeks ($Q_{zf}$), based on the weekly hydrograph and thermograph of the hydrological model. Water temperature is considered the most important physiological threshold for fish species, as mortality of ectothermic species occurs above and below lethal thresholds[8,46]. Decreases in minimum flow directly affect riffle-pool systems and connectivity between viable habitat patches, leading to a rapid loss of biodiversity[47]. We included the number of zero-flow weeks because increases in the frequency of dry-spells directly correlates with reduction in diversity and biomass due to the loss of suitable aquatic habitat[47]. We considered maximum flow because increases in high flow might reduce abundance of young-of-the-year fish by washing away eggs and displacing juveniles and larvae, impeding them from reaching nursery and shelter habitats[47,48].

We quantified species-specific thresholds for minimum and maximum weekly flow, maximum number of zero flow weeks and maximum and minimum weekly water temperature based on the present-day distribution of these characteristics within the geographic range of each species, similarly to previous studies[45,49,50]. To this end, we overlaid the species' range maps with the weekly flow and water temperature metrics from the output of the hydrological model, calculated for each year and averaged over a 30-years historical period to conform to the standard for climate analyses[51,52] (1976–2005, for each GCM employed in the study). We calculated for each 5 arcminutes grid cell the long-term average minimum and maximum weekly flow ($Q_{min}$, $Q_{max}$, Eqs. (1) and (2)), the long-term average frequency of zero-flow weeks ($Q_{zf}$, Eq. (3)) and the long-term average minimum and maximum temperature ($T_{min}$, $T_{max}$, Eqs. (4) and (5)), as follows:

$$Q_{min} = \frac{\sum_{i=1}^{N} \min(Q7_i)}{N} \qquad (1)$$

$$Q_{max} = \frac{\sum_{i=1}^{N} \max(Q7_i)}{N} \qquad (2)$$

$$Q_{zf} = \frac{\sum_{i=1}^{N} \left\{ j \in \{1, \dots, M\} : q7_j = 0 \right\}_i}{N} \qquad (3)$$

$$T_{min} = \frac{\sum_{i=1}^{N} \min(T7_i)}{N} \qquad (4)$$

$$T_{max} = \frac{\sum_{i=1}^{N} \max(T7_i)}{N} \qquad (5)$$

where Q7 and T7 are the vectors of weekly streamflow and water temperature values for a given year $i$, respectively; q7 is the streamflow value for the week $j$; $N$ is the number of years considered (30 in this case) and $M$ is the number of weeks in a year (~52). We then used the spatial distributions of these values within the range of each species to determine species-specific 'thresholds' for each of the variables, defined as the 2.5 percentile of the minimum flow and minimum temperature and the 97.5 percentile of the maximum water temperature and zero flow weeks values. We preferred these to using the absolute minimum and maximum values to reduce the influence of uncertainties and outliers in the threshold definition. Only for maximum flow we used the maximum value across the range, because of the highly right-skewed distribution of flow values within the range of the species. An overview of the thresholds' distribution is available in Supplementary Fig. 8.

**Climate forcing and warming targets**. We considered four main future scenarios based on increases of global mean air temperature equal to 1.5, 2.0, 3.2, and 4.5 °C. The global mean temperature increase refers to a 30-years average, in accordance with guidelines for climate analyses[51], and with pre-industrial reference set at 1850–1900[31]. To obtain estimates of weekly water temperature and flow for each warming level, we forced the hydrological model with the output from an ensemble of five Global Climate Models (GCMs), each run for four Representative Concentration Pathway (RCP) scenarios, namely RCP 2.6, 4.5, 6.0, and 8.5 (see "Supplementary Methods" for details). Hence, each RCP–GCM combination would reach each warming level at a different point in time, with some of the RCP–GCM combinations not reaching certain warming levels. Consequently, the number of scenarios available differed among warming levels (an overview is provided in Supplementary Table 1). In total we modeled 42 scenarios (one scenario = one GCM–RCP combination at a certain point in the future), including 17 scenarios for 1.5 °C, 15 for 2.0 °C, 7 for 3.2 °C and 3 for 4.5 °C.

**Projecting species-specific future climate threats**. For each species and each of the 42 scenarios as described in the previous section, we quantified the proportion of the range where projected extremes exceed the present-day values within the species' range for at least one of the variables. Thus, for each species $x$ we quantified the percentage of geographic range threatened (RT [%]) at each GCM-RCP scenario combination $c$ and for a variable (or group of variables) $v$ as,

$$RT_{x,c,v} = \frac{AT_{x,c,v}}{A_x} \cdot 100 \qquad (6)$$

where AT is the portion of area threatened [km²] and $A$ is the current geographic range size [km²]. That is, we assessed for all grid cells within the species' range if a projected minimum or maximum weekly flow would fall below the minimum or above the maximum flow threshold, if there would be a higher number of zero flow weeks than the threshold would allow, or if the minimum or maximum weekly water temperature would be lower than the minimum or higher than the maximum water temperature threshold. The variable-by-variable evaluation allowed us to identify which (groups of) variable(s) contributed to the threat. For simplicity, we grouped the number of zero flow weeks, minimum and maximum weekly flow variables to assess threat imposed by altered flow regimes. Similarly, we grouped threats imposed by amplified minimum and maximum weekly water temperature to assess temperature-related threats. In the aggregated results, a grid-cell is thus flagged as threatened if any of the underlying thresholds is exceeded.

**Accounting for dispersal**. In general, organisms may adapt to climate change (or escape from future extremes) by moving to more suitable locations[53]. Accounting for this possibility is challenging due to the uncertainties and data gaps associated with current and future barriers in freshwater systems (e.g., dams, weirs, culverts, sluices)[54]. In addition, data needed to reliably estimate dispersal ability is still lacking for the majority of the species[55]. We therefore employed two relatively simple dispersal assumptions in our calculations. Under the "no dispersal" assumption, fishes are restricted to their current geographic range, whereas under the "maximal dispersal" assumption, fishes are assumed to be able to reach any cell within the sub-basin units encompassing their current geograhic range. We defined the sub-basin units by intersecting the physical boundaries of main basins (defined as having an outlet to the sea/internal sink) with the boundaries defined by the freshwater ecoregions of the world, which provide intra-basins divisions based on evolutionary history and additional ecological factors relevant to freshwater fishes[22] (Supplementary Fig. 11). Basins smaller than 1,000 km² were combined with adjacent larger units. In total, we delineated 6,525 sub-basin units (area: $\mu = 20,376$ km², $\sigma = 90,717$ km²) from 10,884 main hydrologic basins and 449 freshwater

ecoregions. To model future climate threats under the maximal dispersal assumption, we first expanded the geographic range for the current situation, allowing the species to occupy grid cells within the encompassing sub-basin boundaries if suitable according to the species-specific thresholds. Then we assessed future climate threats for the 42 different scenarios relative to the present-day range plus all cells potentially available to the species within the encompassing sub-basins (excluding cells that would become threatened in the future), as

$$RT_{x,c,v} = \frac{AT_{x,c,v}}{A_x + (AE_x - AET_{x,c,v})} \cdot 100 \qquad (7)$$

where AE is the expanded part of the geographic range [km²] and AET is the area threatened within the expanded part of the geographic range [km²].

**Aggregation of results**. To summarize our results, we first assessed the proportion of species having more than half of their (expanded) geographic range threatened (i.e., exposed to climate extremes beyond current levels within their range) at each warming level. We did this for each GCM-RCP scenario combination and then calculated the mean and standard deviation across the GCM-RCP combinations at each warming level. We further calculated the proportion of species threatened by future climate extremes in each 5 arcminutes (~10 km at the Equator) grid cell for each warming level, as follows:

$$PAF_{i,w} = median_c \left(1 - \frac{S_{i,w}}{S_{i,present}}\right) \qquad (8)$$

where PAF represents the potentially affected fraction of species in grid cell $i$ for warming level $w$, $c$ represents the scenario (i.e., GCM–RCP combination), $S_{i,w}$ represents the number of species for which extremes in water temperature and flow in grid-cell $i$ according to warming level $w$ do not exceed present-day levels within their range, and $S_{i,present}$ represents the number of species in grid cell $i$. For both numerator and denominator, the species pool for cell $i$ was determined based on the overlap with the (expanded) geographic range maps (see "Species occurrence data" and "Accounting for dispersal"). We used the median across the GCM–RCP combinations rather than the mean because the data showed skewed distributions. Finally, we averaged the grid-specific proportions of species affected over main basins with an outlet to the ocean/sea or internal sink (e.g., lake), as follows:

$$\overline{PAF}_{x,w} = \frac{\sum_{i=1}^{I} PAF_{i,w}}{I_x} \qquad (9)$$

where $I_x$ represents the number of grid cells within the watershed $x$.

**Phylogenetic regression on species traits**. We performed phylogenetic regression to relate the threat level of each species, quantified as the proportion of the geographic range exposed to future climate extremes beyond current levels within the range (see Eqs. (6) and (7)), to a number of potentially relevant species characteristics, while accounting for the non-independence of observations due to phylogenetic relatedness among species[56]. We established a phylogenetic regression model per warming level and dispersal scenario (i.e., eight models in total, based on four warming levels times two dispersal assumptions). As species characteristics, we included initial range size (in km²), body length (in cm), climate zone, trophic group and habitat type, as these traits may influence species' responses to (anthropogenic) environmental change[8,30,57,58]. We further included IUCN Red List category to evaluate the extent to which current threat status is indicative of potential impacts of future climate change, and commercial importance to evaluate implications of potential extirpations for fisheries. We overlaid each species' geographic range with the historic Köppen–Geiger climate categories to obtain the main climate zone per species (i.e., capital letter of the climate classification)[59]. Species falling into multiple climate categories were assigned the climate zone with the largest overlap. We retrieved information on threat status from IUCN[40] and on taxonomy from Fishbase[42]. We used the IUCN and Fishbase data also to gather a list of potential habitats for each species. For the species represented within the IUCN dataset, we classified species as lotic if they were associated with habitats containing at least one of the words "river", "stream", "creek", "canal", "channel", "delta", "estuaries", and as lentic if the habitat descriptions contained at least one of the words "lake","pool","bog","swamp","pond". For the remaining species, we extracted information on habitat from Fishbase, where we used the highest level of aggregation of habitat types to classify species found in lakes as lentic and species found in rivers as lotic. We classified species occurring in both streams and lakes as lotic-lentic and labeled species found in both freshwater and marine environments as lotic-marine. Further, we retrieved data from Fishbase on maximum body length and commercial importance[42]. From the same database we also retrieved trophic level values and aggregated them into Carnivore (trophic level >2.79), Omnivore (2.19 < trophic level ≤ 2.79) and Herbivore (trophic level ≤ 2.19)[42]. We performed a synonym check for the binomial nomenclature provided by the IUCN database to maximize the overlap with the Fishbase database. Since information on phylogeny was available only for a subset of 4,930 fish species covered in our study (based on Betancur-R et al.[60]), we allocated the remaining species to the phylogenetic tree using an imputation procedure implemented in the R package "fishtree"[61]. The empirical tree covered 97% of the families and 80% of the genera included in our species set, indicating that the

majority of the missing species were allocated to the correct genus. Our final dataset for the regression included 9,779 species (695 species were excluded because covariates were not available and 951 because they were not included in the "fishtree" database). To account for the uncertainty in the phylogenetic tree imputation, we repeated each of the eight phylogenetic regression models based on 100 replicates of the phylogenetic tree[61]. Prior to running the regressions we log-transformed threat level (response variable), geographic range size and body length as these variables were right-skewed. As Spearman's rank correlations among the covariates were below 0.4 and variance inflation factors below 1.5 (Supplementary Fig. 12 and Supplementary Table 6), we kept the full set of covariates. We ran the phylogenetic regression using the R package "nlme"[62,63] and checked the residuals of the models using QQ plots (Supplementary Fig. 13). We then extracted coefficients, $|t|$-statistics, $p$ values as well as the lambda parameter at each warming level (averaged over the 100 replicates; Supplementary Table 5). Then, we quantified variable importance using a procedure based on the random forest approach[64], as implemented in the R package "biomod2"[65]. To that end, we randomized the values of the covariates one by one and computed the variable importance as 1 minus the Pearson's $r$ between the predictions of the original model and the predictions obtained from the model with randomized data. We iterated this procedure 10,000 times (100 iterations of the variable importance algorithm times 100 models based on replicates of the phylogenetic tree) and reported the average score and standard deviation across the 100 stochastic replicates (standard deviation across the iterations was negligible) for each of the eight models.

**Reporting summary**. Further information on research design is available in the Nature Research Reporting Summary linked to this article.

## Data availability
Data associated with this publication including source data files for Figs. 1–5 of this manuscript are available within the paper and its supplementary information files. Species' geographical ranges were downloaded from IUCN[40], from Jézéquel et al.[41] and a combination of additional sources as described in Barbarossa et al.[23].

## Code availability
The R code used to model species' threat levels and produce all the figures presented here is available at https://github.com/vbarbarossa/fishsuit[66]. The Python source code used to obtain weekly water temperature and flow estimates at 5 arcminutes is available at https://github.com/UU-Hydro/PCR-GLOBWB_model[67] (PCR-GLOBWB) and at https://github.com/wande001/dynWat (DynWat). All the model runs were carried out on the Dutch national e-infrastructure Cartesius.

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

## Acknowledgements

We would like to thank Felix Leiva for the help provided with the phylogenetic regression and Erin Henry for providing part of the data on critical thermal maxima. This work was carried out on the Dutch national e-infrastructure with the support of SURF Cooperative (account ruc17252). This project has received funding from the Europeans Union's Horizon 2020 research and innovation program under the Marie Sklodowska-Curie grant agreement No. 641459 and the GLOBIO project (www.globio.info). The contribution of M.A.J.H. was financed by the Netherlands Organisation for Scientific Research project no. 016.Vici.170.190.

## Author contributions

V.B. and A.M.S. conceived the idea and wrote the paper. V.B. performed the analyses. J.B. ran the PCR-GLOBWB simulations to produce the hydrological data. V.B., A.M.S., J.B., M.A.J.H., N.W., M.F.P.B., and H.K. contributed to the study design, discussions about the approach and results, and commented on the paper.

## Competing interests

The authors declare no competing interests.
