## [Peer Review File · Nature Communications]

Reviewers' Comments:

Reviewer #1:

Remarks to the Author:

In this study, the authors assessed the risk posed by climate change-induced flow and temperature alterations on the distribution range of freshwater fish at the global scale. The question is timely and will undoubtedly be of interest of many readers. I particularly appreciate the fact that the analyses are based on multiple climate models and scenarios developed at a fine spatial and temporal resolution (although I am not expert enough to be able to assess the potential implications of using only one single hydrologic model). That being said, I think that in its present form, the manuscript is rather descriptive and does not fully exploit the results. For instance, both flow and water temperature alterations are expected to affect freshwater organisms, but their effects are rarely studied in concert. I think that this is a main novelty of the study. The comparison of the magnitude and geography of their effects on freshwater fishes could thus deserve more attention as currently the results are only shown in the supplementary material and superficially discussed. The rationale behind the analyses is also often not properly explained (e.g. selection of traits). Finally, the interpretation and discussion of the results is split between the Results and the Discussion sections, which in my opinion dilute the take home messages of the paper. A more traditional format might be more appropriate and would allow to trim a few generic sentences that do not bring much to the discussion (examples below).

- Whereas I agree that the number of studies devoted to freshwater versus terrestrial or marine systems is highly unbalanced, I think that the Introduction could better acknowledge the climate change-related studies that have been conducted in the freshwater realm, both at the continental (e.g. Thieme et al. 2010), and global (e.g. VanVliet et al. 2013; Comte et al. 2017) scales. Although the specific focus of these studies somewhat differ from what has been conducted here (e.g. they were performed on a subset of species or do not explicitly quantify range contractions), I believe that they are still highly relevant to the present study.

- I am surprised by the lack of effects of predicted hydrologic changes on the distribution range of fish. Perhaps this comes from the selected species-specific thresholds. Related research suggests that many watersheds may experience increased (e.g. northern North America, Asia, central Africa) or decreased (e.g. United States, southern and central Europe) discharge in the future (van Vliet et al. 2013, 2016, Thieme et al. 2010) and the results presented in Fig. 4 seem to confirm that important changes are expected in the two directions. However, only the effects of decreased flow are considered here, whereas the effects of dampened flow seasonality or increased flows may be consequential on species whose life histories are adapted to the local flow regime (Bunn & Arthington, 2002). Also, how the lentic species were considered in this analysis, were they excluded or were alternative thresholds more relevant to lakes considered? I appreciate the Supplementary Discussion about the difficulty of selecting biologically relevant thresholds, but I think that these methodological steps deserve more justifications and constructive criticisms.

- By computing empirical temperature and flow thresholds, the authors implicitly assume that all species are at equilibrium with their environments, which may not be the case and in turn may affect our understanding of geographical patterns of climate vulnerability (Sunday et al. 2012). For instance, the fact that species spending part of their life in the ocean are predicted to be the least affected may reflect biases in the estimation of their sensitivity to climate change (i.e. truncated niches). The same may apply to lentic species but in the opposite direction (which is acknowledged in the results section). Although the authors compared their metric of temperature sensitivity to laboratory experiments (L51-53), they did it for only 213 species out of the 6924 included in the analyses, so I am not sure that this could be considered as a validation of the methodology. I understand that it is impossible to account for all the factors affecting species distribution ranges in global-scale analyses but I would like to see more discussions about this potential issue.

Specific comments

L70-75 Already stated in Introduction. Could be easily shortened

L108-109 Why using a reference to describe the predicted patterns of flow alteration and temperature? Is it not an original result of this study? A comparison with previous research would be more useful. The color scale in Supplementary Fig 4 is also difficult to read. For instance, on Fig 4-III it is almost impossible to distinguish the different colors. The color scale varies between - 30 and 30 degC, does it mean that certain areas are expected to see an increase in water temperature of 30degC?? I have the same comment for the Fig. 4-II where the scale for the number of zero flow weeks varies between -40 and 40. Perhaps using another color scale or an uneven scale (on Fig 4-I very few pixels are not grey) would help. Alternatively, showing the % of changes would be more informative(?).

L84 change recorded by predicted

L110-113 Why a reference to salmonids here? The fact that climate change is expected to threaten fish species and that more studies are needed have already been explained in Introduction (especially not using a reference from 2012 whereas many studies have since been published).

L114 Comparison across species groups and traits

Potential differences among groups are interpreted based on visual comparisons presented in Fig. 3. It would be interesting to formally test these differences and provide statistics. Also, I would appreciate an explanation of why these particular traits were selected for the analyses (theoretical context and predictions), and how the groups for the quantitative traits (e.g. body size) presented in Fig. 3 were defined.

Supplementary Fig. 3: I am not sure to understand what is represented on panel (c) as the relative loss is lower than in panel (b). Shouldn't it represent (a) + (b)?

Cited references

Bunn, S. E., & Arthington, A. H. (2002). Basic principles and ecological consequences of altered flow regimes for aquatic biodiversity. *Environmental Management*, 30, 492–507.

Comte, L. & Olden, J. D. Climatic vulnerability of the world's freshwater and marine fishes. *Nat. 297 Clim. Chang.* 7:718–722 (2017).

Sunday, J. M., Bates, A. E., & Dulvy, N. K. (2012). Thermal tolerance and the global redistribution of animals. *Nature Climate Change*, 2, 686–690.

Thieme ML, Lehner B, Abell R, Matthews J. 2010. Exposure of Africa's freshwater biodiversity to a changing climate. *Conserv. Lett.* 3:324–31

van Vliet MT, Ludwig F, Kabat P. 2013 Global streamflow and thermal habitats of freshwater fishes under climate change. *Clim. Change* 121:739–54

van Vliet MT, D Wiberg, S Leduc, K Riahi 2016 Power-generation system vulnerability and adaptation to changes in climate and water resources. *Nature Climate Change* 6, 375.

Reviewer #2:

Remarks to the Author:

Review of NCOMMS-19-17479

In this manuscript, Barbarossa et al. quantified potential range contractions of freshwater fish globally under various warming scenarios. Importantly, the authors also compared the effects of two facets of climate change-induced stressors (high water temperature and low flow) on range contractions.

The topic of study is important and of general interest to both climate change scientists, conservation ecologists, and the public. However, there are several issues that may limit the impact and relevance of this study. I summarize these issues below.

First, the most important limitation is that the study only quantified the area lost from a species' current range. I understand the limitations with species dispersal and barrier datasets; they are currently quite scattered and would need to be carefully compiled. I also appreciate the authors' explanation why they did not model range expansion. Nevertheless, to truly understand changes in species ranges, I think it is crucial to incorporate both loss of suitable habitat in current ranges as well as potential expansion to new areas in the analysis.

Second, it is unclear whether hydrological indices at the relatively coarse 10 km grain are appropriate for modeling fish distributions. Flow is an important predictor of habitat suitability at the scale of individual stream reaches. At the scale of a 10 km cell, flow is aggregated and therefore affected by the density of water bodies. Flow conditions in each water body would only be comparable if the density of water bodies is roughly similar across grid cells. Some discussion on the implications of the scale of analysis on species range projections is needed.

Third, I suggest the authors perform formal statistical analyses to examine the relationship between range contractions and body size, initial range size, habitat types, etc.

Last, I think the authors should describe their methods in greater detail. For example, it is unclear how the loss of species ranges due to water temperature vs. low flow was calculated (Supplementary Fig. 3), and there is no description on how the two original flow metrics (minimum weekly flow and number of zero flow weeks) are combined into a single "low flow" variable. I presume that the bioclimatic envelope model was applied using a single water temperature variable to calculate range losses due to water temperature. To calculate range losses based on flow, I assume the bioclimatic envelope used both flow variables. If that is the case, I suggest the authors to state the approach explicitly in the Methods section. Readers can also benefit from having more background information on the GHM and the water temperature model and how these models are coupled. In addition, a description of the bioclimatic envelope modeling procedure should be presented in the Methods section.

I hope my comments and suggestions are useful for the authors.

Specific comments

L42. 6,924 out of ~15,000 fish species were analyzed based on availability of range polygon data from IUCN. Could there be differences between species that are included in the IUCN database versus those that are excluded that may affect the generality of the conclusions?

L53 and Supplementary Fig 1. It is unclear how mean standard error describes the relationship between species-specific water temperature thresholds and lab-measured critical thermal maxima. I suggest the authors explain this metric in the Methods section.

L53-54. The bioclimatic envelope model approach should be described here, otherwise it is unclear how changes in species ranges were calculated.

L130. Insert "species to be" between "non-migratory" and "the most affected".

L134. Replace "a good predictor" with "good predictors".

L157. "retaining ~60%". Do the authors mean that 60% of the fish species analyzed in this study are found in tropical areas? I suggest rephrasing this sentence or breaking it down to two sentences.

Supplementary Fig 8. I suggest plotting watersheds with no data in grey so that it is consistent with Supp. Fig. 7 (pixels with no data are colored gray).

Methods. L193. The IUCN spatial data are more accurately described as species ranges (or even more accurately, species range polygons) instead of occurrences.

Methods. L196-198. "We referenced the occurrence data...of the species". This sentence is not very clear. Are the species range polygons resampled to a ~10 km grid, with a given species marked as occurring in a cell if $\geq 50\%$ of the cell area overlapped with the species range polygon?

N.B. Line numbers refer to the clean version of the manuscript (not the one with highlighted changes)

Reviewer #1 (Remarks to the Author)

1) In this study, the authors assessed the risk posed by climate change-induced flow and temperature alterations on the distribution range of freshwater fish at the global scale. The question is timely and will undoubtedly be of interest of many readers. I particularly appreciate the fact that the analyses are based on multiple climate models and scenarios developed at a fine spatial and temporal resolution (although I am not expert enough to be able to assess the potential implications of using only one single hydrologic model). That being said, I think that in its present form, the manuscript is rather descriptive and does not fully exploit the results. For instance, both flow and water temperature alterations are expected to affect freshwater organisms, but their effects are rarely studied in concert. I think that this is a main novelty of the study. The comparison of the magnitude and geography of their effects on freshwater fishes could thus deserve more attention as currently the results are only shown in the supplementary material and superficially discussed. The rationale behind the analyses is also often not properly explained (e.g. selection of traits). Finally, the interpretation and discussion of the results is split between the Results and the Discussion sections, which in my opinion dilute the take home messages of the paper. A more traditional format might be more appropriate and would allow to trim a few generic sentences that do not bring much to the discussion (examples below).

Thank you for your kind words. We have now moved the figure showing the contribution of water temperature vs flow from the SI to the main text and accordingly enriched the corresponding results description and discussion. Further, we have added more explanation on the rationale behind our analyses, including the dispersal scenarios (L. 289-299), the selection of flow metrics (L. 54-57 and L. 146-157) and the selection of species traits (see further our reply to comment 9). Finally, we moved all the discussion elements to the discussion section, as we agree that a more traditional format better brings across the messages of our study.

2) Whereas I agree that the number of studies devoted to freshwater versus terrestrial or marine systems is highly unbalanced, I think that the Introduction could better acknowledge the climate change-related studies that have been conducted in the freshwater realm, both at the continental (e.g. Thieme et al. 2010), and global (e.g. VanVliet et al. 2013; Comte et al. 2017) scales. Although the specific focus of these studies somewhat differ from what has been conducted here (e.g. they were performed on a subset of species or do not explicitly quantify range contractions), I believe that they are still highly relevant to the present study.

As recommended, we now more explicitly acknowledge existing studies quantifying impacts of climate change on freshwater fish species in the introduction section. We added the following sentence with references to the suggested studies at L. 41-42 “Recent continental and global studies have underscored the high vulnerability of freshwater fish species to climate change^{8,11-13}.”

3) I am surprised by the lack of effects of predicted hydrologic changes on the distribution range of fish. Perhaps this comes from the selected species-specific thresholds. Related research suggests that many watersheds may experience increased (e.g. northern North America, Asia, central Africa) or decreased (e.g. United States, southern and central Europe) discharge in the future (van Vliet et al. 2013, 2016, Thieme et al. 2010) and the results presented in Fig. 4 seem to confirm that important changes are

expected in the two directions. However, only the effects of decreased flow are considered here, whereas the effects of dampened flow seasonality or increased flows may be consequential on species whose life histories are adapted to the local flow regime (Bunn & Arthington, 2002). Also, how the lentic species were considered in this analysis, were they excluded or were alternative thresholds more relevant to lakes considered? I appreciate the Supplementary Discussion about the difficulty of selecting biologically relevant thresholds, but I think that these methodological steps deserve more justifications and constructive criticisms.

Thank you for your comment. Indeed, according to our results, effects of changes in water temperature are expected to considerably exceed those of altered flow (new Figure 4 in the main text). We now elaborate in the discussion on the relatively small effect of changes in flow, as follows (L. 146-157): “Our results suggest that future increases in water temperature have larger impacts on freshwater fish species distributions than changes in low flow conditions. In line with previous studies, we found that climate change will result in reduced flows mainly in drought-prone regions, while many others will experience an increase in flow^{20,28}. In addition, depletion of low flows might be most important at low stream orders, which are not well captured by the 5 arcminutes resolution of the hydrological model PCR-GLOBWB employed in this study¹⁹. Hence, the spatial resolution of the analysis might result in an underestimation of the impacts of climate change on species living in smaller upstream reaches. While global estimates of hydrological variables are available at higher spatial resolutions²⁹, 5 arcminutes is the highest resolution achievable so far for future projections of both flow and water temperature²⁰. We further note that we only considered critical low flow metrics in our study^{30,31}, thus ignoring potential range losses due to changes in flow seasonality or increased flows, which might disfavor species whose life histories are adapted to the local flow regime^{5,32}.”

Further, we acknowledge the caveats of using the same thresholds for lentic species, as follows (L. 177-180): “It should be noted, however, that our range loss estimates for species living in lakes with large bathymetric ranges are likely to be overestimated. Since our hydrological model provides only average water temperature estimates in lakes²⁰, we could not account for microclimatic refugia offered by water stratification³⁶.”

4) By computing empirical temperature and flow thresholds, the authors implicitly assume that all species are at equilibrium with their environments, which may not be the case and in turn may affect our understanding of geographical patterns of climate vulnerability (Sunday et al. 2012). For instance, the fact that species spending part of their life in the ocean are predicted to be the least affected may reflect biases in the estimation of their sensitivity to climate change (i.e. truncated niches). The same may apply to lentic species but in the opposite direction (which is acknowledged in the results section). Although the authors compared their metric of temperature sensitivity to laboratory experiments (L51-53), they did it for only 213 species out of the 6924 included in the analyses, so I am not sure that this could be considered as a validation of the methodology. I understand that it is impossible to account for all the factors affecting species distribution ranges in global-scale analyses but I would like to see more discussions about this potential issue.

We agree and now more extensively reflect on the implications of assuming range equilibrium in the discussion, as follows (L. 159-170): “Impacts of temperature change might, however, be overestimated due to the intrinsic assumption that species’ current distributions are confined by water temperature, which might not be the case everywhere. For example, species ranges might also be influenced by biogeographic dispersal barriers or anthropogenic pressures, implying that species might be able to

occupy a wider range of environmental temperatures than assumed based on their range maps^{33,34}. Yet, our tentative comparison of the empirical maximum weekly water temperature thresholds against critical thermal maxima determined in the lab, showed no systematic underestimation (Supplementary Figure 1), possibly reflecting that the geographic ranges of aquatic species may conform more closely to their limits of thermal tolerance than the ranges of terrestrial species³⁴. Further work is required to better understand deviations between our environmental thresholds and the lab-based maxima, which may at least partly reflect heterogeneity in experimental conditions³⁵.”

5) L70-75 Already stated in Introduction. Could be easily shortened

We agree. We removed those lines from the Results sections and used part of it in the Discussion section.

6) L108-109 Why using a reference to describe the predicted patterns of flow alteration and temperature? Is it not an original result of this study? A comparison with previous research would be more useful. The color scale in Supplementary Fig 4 is also difficult to read. For instance, on Fig 4-III it is almost impossible to distinguish the different colors. The color scale varies between – 30 and 30 degC, does it mean that certain areas are expected to see an increase in water temperature of 30degC?? I have the same comment for the Fig. 4-II where the scale for the number of zero flow weeks varies between -40 and 40. Perhaps using another color scale or an uneven scale (on Fig 4-I very few pixels are not grey) would help. Alternatively, showing the % of changes would be more informative(?).

Thank you for your comment. The patterns of flow and water temperature are indeed a result of this study. We now make this more explicit in the results section, at L. 103-106: “This reflects that climate change is projected to result in more severe low flow conditions mainly in drought-prone regions, such as the Mediterranean, Sub-Saharan Africa, parts of the US, South America and Australia, while water temperature rises almost everywhere (Supplementary Figure 4-III).” We then compare the similarity of patterns to existing literature only in the discussion section, at L. 147-149 as follows “In line with previous studies, we found that climate change will result in reduced flows mainly in drought-prone regions, while many others will experience an increase in flow^{20,28}.” In addition, we now show the results in Supplementary Figure 4-I,II,III in terms of percentage of change rather than absolute values and adjusted the color scale accordingly.

7) L84 change recorded by predicted

We changed this accordingly.

8) L110-113 Why a reference to salmonids here? The fact that climate change is expected to threaten fish species and that more studies are needed have already been explained in Introduction (especially not using a reference from 2012 whereas many studies have since been published).

We agree and have removed that sentence.

9) L114 Comparison across species groups and traits

Potential differences among groups are interpreted based on visual comparisons presented in Fig. 3. It would be interesting to formally test these differences and provide statistics. Also, I would appreciate an explanation of why these particular traits were selected for the analyses (theoretical context and predictions), and how the groups for the quantitative traits (e.g. body size) presented in Fig. 3 were defined.

Thank you for this valuable suggestion. We now more formally test the differences among species groups by performing a phylogenetic regression analysis relating the climate-induced range changes to the species traits. We revised our selection of traits and removed the grouping for the continuous traits. We inserted a new section in the methods (L. 306-341) detailing all the steps used for the phylogenetic regression and explaining the rationale behind the choice of the traits, as follows: “We performed a regression analysis between projected geographical range changes and a number of species characteristics. Since observation at the species level are not independent, we used phylogenetic regression to account for phylogenetically close species that tend to be similar due to their evolutionary history⁴⁷. Data on phylogeny was available for a subset of the fish species employed in our study from Betancur-R et al.⁴⁸, and therefore the phylogenetic regression analysis was performed on a sample of 2,757 species. As species characteristics, we included initial range size (in km²), body length (in cm), climate zone, trophic group and habitat type, as these traits may influence species’ responses to (anthropogenic) environmental change^{8,35,49,50}. We further included IUCN Red List category to evaluate the extent to which current threat status is indicative of impacts of future climate change, and commercial importance to evaluate potential implications of future range changes for fisheries. We overlaid each species occurrence range with the historic Köppen-Geiger climate categories to obtain the main climate zone per species (i.e., capital letter of the climate classification)⁵¹. Species falling into multiple climate categories were assigned the climate zone with the largest overlap. We retrieved information on threat status and species order directly from the IUCN metadata⁴¹. We also gathered a list of potential habitats for each species from the same IUCN database. We classified species as lotic if they were associated with habitats containing at least one of the words “river”, “stream”, “creek”, “canal”, “channel”, “delta”, “estuaries”, and as lentic if the habitat descriptions contained at least one of the words “lake”, “pool”, “bog”, “swamp”, “pond”. Further, we retrieved data from Fishbase (www.fishbase.org) on maximum body length and commercial importance⁵². From the same database we also retrieved trophic level values and aggregated them into Carnivore (trophic level >2.79), Omnivore (2.19 < trophic level ≤ 2.79) and Herbivore (trophic level ≤ 2.19)⁵². We performed a synonym check for the binomial nomenclature provided by in the IUCN database to maximize the overlap with the Fishbase database. Prior to running the regression we log-transformed range area and body length as these variables were highly skewed. Then we checked the bivariate Pearson correlations among the species characteristics, which were mostly below 0.5 (Supplementary Figure 10). Finally, we calibrated the lambda coefficient of the phylogenetic regression using the R package “nlme”^{53,54}. We calibrated the lambda parameter for the geographical range changes at the different warming levels and for the different dispersal assumptions. We used a final lambda value of 0.96, which is the average calibrated value across the different response variables. Using the final regression models, we extracted the coefficients and quantified variable importance using a statistical procedure that randomizes the values of the covariate of concern and then computes the variable importance as 1 minus the Pearson’s r,

based on the original model against the prediction obtained from the randomized data⁵⁵. We iterated this 100 times for each variable of the eight models (four warming levels times two dispersal assumptions).”

10) Supplementary Fig. 3: I am not sure to understand what is represented on panel (c) as the relative loss is lower than in panel (b). Shouldn't it represent (a) + (b)?

Thank you for your comment. We have now moved the figure to the main text (new Figure 4) and made more explicit what each panel means in the figure caption, as follows: “Figure 4. Contributions of increased water temperature and more severe low flow conditions to aggregated area losses (AL) across the freshwater fish species for each five arcminutes grid-cell at 3.2°C. Water temperature shows the AL due to the exceedance of the maximum weekly water temperature thresholds, while the low flow groups together the maximum number of zero flow weeks and the minimum weekly flow exceedance thresholds. The LCRL due to exceedance of both flow and water temperature thresholds is reported at the bottom. The maps represent the median AL of the GCM-RCP ensemble available for the 3.2°C warming scenario. Results for the other warming levels are available as Supplementary Figure 3. Gray denotes no data areas (no species occurring or no data available).”

Cited references

- Bunn, S. E., & Arthington, A. H. (2002). Basic principles and ecological consequences of altered flow regimes for aquatic biodiversity. *Environmental Management*, 30, 492–507.
- Comte, L. & Olden, J. D. Climatic vulnerability of the world's freshwater and marine fishes. *Nat. Clim. Chang.* 7:718–722 (2017).
- Sunday, J. M., Bates, A. E., & Dulvy, N. K. (2012). Thermal tolerance and the global redistribution of animals. *Nature Climate Change*, 2, 686–690.
- Thieme ML, Lehner B, Abell R, Matthews J. 2010. Exposure of Africa's freshwater biodiversity to a changing climate. *Conserv. Lett.* 3:324–31
- Van Vliet MT, Ludwig F, Kabat P. 2013 Global streamflow and thermal habitats of freshwater fishes under climate change. *Clim. Change* 121:739–54.
- Van Vliet MT, D Wiberg, S Leduc, K Riahi 2016 Power-generation system vulnerability and adaptation to changes in climate and water resources. *Nature Climate Change* 6, 375.

N.B. Line numbers refer to the clean version of the manuscript (not the one with highlighted changes)

Reviewer #2 (Remarks to the Author)

1) In this manuscript, Barbarossa et al. quantified potential range contractions of freshwater fish globally under various warming scenarios. Importantly, the authors also compared the effects of two facets of climate change-induced stressors (high water temperature and low flow) on range contractions. The topic of study is important and of general interest to both climate change scientists, conservation ecologists, and the public. However, there are several issues that may limit the impact and relevance of this study. I summarize these issues below.

First, the most important limitation is that the study only quantified the area lost from a species' current range. I understand the limitations with species dispersal and barrier datasets; they are currently quite scattered and would need to be carefully compiled. I also appreciate the authors' explanation why they did not model range expansion. Nevertheless, to truly understand changes in species ranges, I think it is crucial to incorporate both loss of suitable habitat in current ranges as well as potential expansion to new areas in the analysis.

Thank you for raising this point. We agree and have revisited our modelling strategy by incorporating a "maximal dispersal" scenario, as follows (L. 290-299): "We employed two dispersal assumption to model geographical range changes, representing two extreme scenarios. Under the no dispersal assumption, fish are restricted to their current geographical range and therefore ranges cannot shift. Under the maximal dispersal assumption, fish are allowed to move anywhere outside their range but within the boundaries of the encompassing watersheds. For instance, a fish found in the Rio Negro, Brazil, would be able to move anywhere within the Amazon basin, but could not cross the watershed boundary to the Parana River basin. To model range changes under this assumption, we first expanded the geographical range for the current situation, allowing the species to occupy grid cells within the watershed boundaries if suitable according to the species-specific thresholds. Then we projected future range changes for the 42 different scenarios in the same way we did for the no dispersal assumption, yet based on all the cells within the watersheds." We have reshaped the text and figures throughout to accommodate the results from both the no dispersal scenario (the previous version of the manuscript) and the new maximal dispersal assumption. We find that spatial patterns of range loss differ among the scenarios (Figure 2) and that average range losses are reduced by 13 to 38% when assuming maximal dispersal when compared to no dispersal.

2) Second, it is unclear whether hydrological indices at the relatively coarse 10 km grain are appropriate for modeling fish distributions. Flow is an important predictor of habitat suitability at the scale of individual stream reaches. At the scale of a 10 km cell, flow is aggregated and therefore affected by the density of water bodies. Flow conditions in each water body would only be comparable if the density of water bodies is roughly similar across grid cells. Some discussion on the implications of the scale of analysis on species range projections is needed.

We agree and addressed this issue by elaborating on the implications of the 5 arcminutes spatial resolution in terms of flow changes at L. 149-154 (see also our reply to comment 3 by reviewer 1), as follows: "In addition, depletion of low flows might be most important at low stream orders, which are not well captured by the 5 arcminutes resolution of the hydrological model PCR-GLOBWB employed in this study¹⁹. Hence, the spatial resolution of the analysis might result in an underestimation of the

impacts of climate change on species living in smaller upstream reaches. While global estimates of hydrological variables are available at higher spatial resolutions²⁹, 5 arcminutes is the highest resolution achievable so far for future projections of both flow and water temperature²⁰.”

3) Third, I suggest the authors perform formal statistical analyses to examine the relationship between range contractions and body size, initial range size, habitat types, etc.

Thanks for this suggestion. As recommended, we now perform a formal statistical analysis by implementing a phylogenetic regression between range changes and species traits. We inserted a new section in the end of the methods, detailing all the steps used for the phylogenetic regression and the rationale behind the choice of the predictors. At L. 306-341, “We performed a regression analysis between projected geographical range changes and a number of species characteristics. Since observation at the species level are not independent, we used phylogenetic regression to account for phylogenetically close species that tend to be similar due to their evolutionary history⁴⁷. Data on phylogeny was available for a subset of the fish species employed in our study from Betancur-R et al.⁴⁸, and therefore the phylogenetic regression analysis was performed on a sample of 2,757 species. As species characteristics, we included initial range size (in km²), body length (in cm), climate zone, trophic group and habitat type, as these traits may influence species’ responses to (anthropogenic) environmental change^{8,35,49,50}. We further included IUCN Red List category to evaluate the extent to which current threat status is indicative of impacts of future climate change, and commercial importance to evaluate potential implications of future range changes for fisheries. We overlaid each species occurrence range with the historic Köppen-Geiger climate categories to obtain the main climate zone per species (i.e., capital letter of the climate classification)⁵¹. Species falling into multiple climate categories were assigned the climate zone with the largest overlap. We retrieved information on threat status and species order directly from the IUCN metadata⁴¹. We also gathered a list of potential habitats for each species from the same IUCN database. We classified species as lotic if they were associated with habitats containing at least one of the words “river”, “stream”, “creek”, “canal”, “channel”, “delta”, “estuaries”, and as lentic if the habitat descriptions contained at least one of the words “lake”, “pool”, “bog”, “swamp”, “pond”. Further, we retrieved data from Fishbase (www.fishbase.org) on maximum body length and commercial importance⁵². From the same database we also retrieved trophic level values and aggregated them into Carnivore (trophic level >2.79), Omnivore (2.19 < trophic level ≤ 2.79) and Herbivore (trophic level ≤ 2.19)⁵². We performed a synonym check for the binomial nomenclature provided by in the IUCN database to maximize the overlap with the Fishbase database. Prior to running the regression we log-transformed range area and body length as these variables were highly skewed. Then we checked the bivariate Pearson correlations among the species characteristics, which were mostly below 0.5 (Supplementary Figure 10). Finally, we calibrated the lambda coefficient of the phylogenetic regression using the R package “nlme”^{53,54}. We calibrated the lambda parameter for the geographical range changes at the different warming levels and for the different dispersal assumptions. We used a final lambda value of 0.96, which is the average calibrated value across the different response variables. Using the final regression models, we extracted the coefficients and quantified variable importance using a statistical procedure that randomizes the values of the covariate of concern and then computes the variable importance as 1 minus the Pearson’s r, based on the original model against the prediction obtained from the randomized data⁵⁵. We iterated this 100 times for each variable of the eight models (four warming levels times two dispersal assumptions).” The results of the regression are presented in Figure 6 (variable importance), Supplementary Table 3 (regression coefficients) and corresponding text in the results description (L. 111-133) and discussion (L. 171-182).

4) Last, I think the authors should describe their methods in greater detail. For example, it is unclear how the loss of species ranges due to water temperature vs. low flow was calculated (Supplementary Fig. 3), and there is no description on how the two original flow metrics (minimum weekly flow and number of zero flow weeks) are combined into a single “low flow” variable. I presume that the bioclimatic envelope model was applied using a single water temperature variable to calculate range losses due to water temperature. To calculate range losses based on flow, I assume the bioclimatic envelope used both flow variables. If that is the case, I suggest the authors to state the approach explicitly in the Methods section. Readers can also benefit from having more background information on the GHM and the water temperature model and how these models are coupled. In addition, a description of the bioclimatic envelope modeling procedure should be presented in the Methods section.

Thank you for these suggestions. We now provide more methodological details on the bioclimatic envelope modelling by compiling a new section “Modelling species-specific geographical range changes” (L. 279-288), where we explain more in detail the bioclimatic envelope modelling and assessment of the contributions of the different hydrological variables, as follows: “For each species, we modelled changes in its geographical range for each of the 42 scenarios as described in the previous section. We modelled range changes by flagging the 5 arc minutes cells as unsuitable when the projected values of flow or water temperature would fall beyond any of the species-specific thresholds (that is, if the minimum weekly flow would fall below the low flow threshold, if there would be a higher number of zero flow weeks than the threshold would allow, or if the maximum weekly water temperature would be higher than the maximum water temperature threshold). The threshold-specific approach allowed us to identify which variable contributed to the unsuitability of the grid cell. For simplicity, when presenting the results, we group the number of zero flow weeks and minimum weekly flow variables together. In this resulting “flow” variable, a grid-cell is considered unsuitable if any of the two mentioned thresholds are exceeded.” In addition, we provide more background information on the GHM and water temperature model in the methods, L. 226-242, as follows: “We employed the Global Hydrological Model (GHM) PCR-GLOBWB¹⁹ with a full dynamical two-way coupling to the Dynamical Water temperature model (DynWAT)²⁰ at 5 arcminutes spatial resolution (~10 km at the equator), to retrieve weekly streamflow and water temperature worldwide^{19,20}. PCR-GLOBWB simulates the vertical water balance between two soil layers and a groundwater layer, with up to four land cover types considered per grid cell. Surface runoff, interflow and groundwater discharge are routed along the river network using the kinematic wave approximation of the Saint-Venant Equations²⁰ and includes floodplain inundation. Apart from the larger lakes, PCR-GLOBWB includes over 6,000 man-made reservoirs⁴³ as well as the effects of water use for irrigation, livestock, domestic and industrial sectors. River discharge, river and lake water levels, surface water levels, runoff fluxes (surface runoff, interflow and groundwater discharge) and water abstractions. The fluxes from PCR-GLOBWB are dynamically coupled to DynWAT along with the meteorological forcing, such as air temperature and radiation, from the GCMs to compute water temperature. DynWAT thus includes temperature advection, radiation and sensible heating but also ice formation and breakup, thermal mixing and stratification in large water bodies, effects of water abstraction and reservoir operations. We selected this model combination because it allows a full representation of the hydrological cycle (considering also anthropogenic stressors, e.g. water use), it fully integrates water temperature and calculates the hydrological variables on a high-resolution hydrography.”

5) L42. 6,924 out of ~15,000 fish species were analyzed based on availability of range polygon data from IUCN. Could there be differences between species that are included in the IUCN database versus those that are excluded that may affect the generality of the conclusions?

Thank you for your comment. Indeed the IUCN database might be biased to well-known species. We identify regions with higher potential bias in northern Asia, South America, and Australia and indicate possible implications of this bias for the generality of our results, see L. 188-195 “Further, the spatial patterns of range loss as found in our study are contingent on the availability of species occurrence data (Supplementary Figure 7). For some basins in northern Asia, South America, and Australia, the fish species range maps available covered less than 20% of the known occurring species² (Supplementary Figure 8). This introduces uncertainty in our results particularly if the missing species have different characteristics than the sample of species included in our study. For example, our aggregated range losses might be underestimated if the IUCN database is biased to wide-ranging species, which according to our regression model are less prone to future climate-induced range loss (Supplementary Table 3).”

6) L53 and Supplementary Fig 1. It is unclear how mean standard error describes the relationship between species-specific water temperature thresholds and lab-measured critical thermal maxima. I suggest the authors explain this metric in the Methods section.

Thank you for your comment. In the supplementary methods section “Validation of the water temperature threshold” we now provide more details on the metrics used to express the goodness of fit between critical thermal limits determined by lab experiments and those inferred in our study, as follows: “To test the goodness of fit between inferred and lab data, we calculated the coefficient of determination R^2 and the mean standard error as $\sum_N \frac{|CT_{max_{lab}} - CT_{max_{inferred}}|}{CT_{max_{lab}}} / N$.”

7) L53-54. The bioclimatic envelope model approach should be described here, otherwise it is unclear how changes in species ranges were calculated.

As recommended, we now describe in more detail the bioclimatic envelope approach in the introduction at L. 54-66, as follows: “To assess range changes, we first compiled species-specific thresholds for three key habitat factors determining the distribution of freshwater fish species²¹, namely maximum weekly water temperature, minimum weekly flow and the number of zero flow weeks, based on extreme flow and water temperature conditions in the species' geographical ranges (see Methods). We compared the species-specific water temperature thresholds with critical thermal maxima reported from laboratory tests⁸ and found them in good agreement (mean standard error = 0.09 [-]; Supplementary Figure 1). We then used the thresholds to quantify changes in geographical ranges due to future changes in flow and thermal regimes. To that end, we delineated for each warming scenario, which parts of the geographical range of each species would become unsuitable due to projected water temperature or flow conditions exceeding at least one of the corresponding thresholds. We did this for two dispersal assumptions: ‘no dispersal’ assuming that each fish species cannot move outside its current geographical range, and ‘maximum dispersal’ assuming that each fish species can move anywhere within the watersheds encompassing its geographical range.”

8) L130. Insert “species to be” between “non-migratory” and “the most affected”.

Corrected, as suggested.

9) L134. Replace “a good predictor” with “good predictors”.

Corrected, as suggested.

10) L157. “retaining ~60%”. Do the authors mean that 60% of the fish species analyzed in this study are found in tropical areas? I suggest rephrasing this sentence or breaking it down to two sentences.

We deleted the statement when rephrasing that paragraph.

11) Supplementary Fig 8. I suggest plotting watersheds with no data in grey so that it is consistent with Supp. Fig. 7 (pixels with no data are colored gray).

Thank you for your suggestion. We have adjusted the figure accordingly.

12) Methods. L193. The IUCN spatial data are more accurately described as species ranges (or even more accurately, species range polygons) instead of occurrences.

Corrected, as suggested.

13) Methods. L196-198. “We referenced the occurrence data...of the species”. This sentence is not very clear. Are the species range polygons resampled to a ~10 km grid, with a given species marked as occurring in a cell if $\geq 50\%$ of the cell area overlapped with the species range polygon?

We agree and rephrased the sentence at L. 220-222 as follows: “We resampled the polygons of each species to the 5 arcminutes (~10 km) hydrography of the global hydrological model (see below), with a given species marked as occurring in a cell if $\geq 50\%$ of the cell area overlapped with the species’ polygon.”

Reviewers' Comments:

Reviewer #1:

Remarks to the Author:

This is the second time that I review the study from Barbarossa et al. and I have to say that it seems to be an entirely different manuscript. The presentation and the content are much improved, and the manuscript reads very well. I also appreciate the detailed responses to my previous comments and the fact that several limits of the study are now explicitly discussed in the main text. I think that the strength of the present study is to provide future water temperature and flow projections under a variety of scenarios; data that are so much needed (but too often lacking) to assess the vulnerability of freshwater organisms to climate change. However, I still have major reserves regarding the study design and the ecological interpretation of the results. The ecological metrics are very simplistic and some of the underlying assumption ecologically questionable. Although this study undoubtedly represents a useful starting point for future investigations, I am worried that the authors are trying to say too much based on their results, without taking the time to carefully consider the underlying assumptions and limits of their approach.

Terminology

First of all, I think that there is a confusion regarding the methodology that was used to estimate future range changes. The approach taken is fairly simplistic by considering that areas projected to be above/below the maximum or minimum water temperature and streamflow currently observed across species ranges will be unsuitable for the species in the future, resulting in range contractions. Conversely, the future expansion scenarios consider that species can expand in any grid cell present in the basin if the conditions stay below the previously identified thresholds. Given the approach I think that the terminology used in the manuscript (e.g. 'Modelling species-specific geographical range changes' (L278), 'we employed two dispersal assumption to model geographical range changes' (L290), 'we projected future range changes' L298, 'our model projections' L211) is misleading as it suggests that some sort of bioclimatic modelling approaches were used. I also question the comparison with the study of Warren et al. 2018 for terrestrial vertebrates (L205-207; also L302) as the later study did use modelling approaches (i.e. Maxent) to estimate future extirpation risk. Still related to this issue, I do not think that this study provides a 'comprehensive quantification of potential geographical range changes of freshwater fish in relation to future climate change' as claimed L135. In my opinion this study is not comprehensive either in terms of taxonomic coverage (~7000 species) or future climate change effects on geographical range (only maximum water temperature and minimum streamflow were considered). I thus suggest to lower the tone and reconsider the terminology used throughout the manuscript.

Ecological assumptions

The ecological assumptions behind the approach are also questionable as they assume that maximum weekly water temperature, minimum weekly flow and the number of zero flow weeks are the only factors affecting the distribution of freshwater fishes and that they act independently from each other. Although the approach can be defensible regarding future extirpation risk (though many caveats must be acknowledged as I highlighted in my previous round of comments), I believe that the resulting assumptions for the dispersal scenarios are ecologically untenable. For instance, species can disperse to any given pixel as defined by the maximum weekly temperature predicted in the future with no consideration whatsoever for hydrological or other habitat conditions. I understand that the authors included an estimation of future range expansion in an attempt to respond to the concerns raised by the reviewer #2, but I don't think that the approach taken is the right answer. If the goal of the study is to assess potential changes in habitat suitability in the future, then the authors may want to consider using species distribution models. I do not want to imply here that these models are perfect tools but they have at least the merit to account for the multidimensional nature of species niches.

Estimation of habitat suitability thresholds

I also still have reserves regarding the estimation of habitat suitability thresholds. These were derived using empirical data defined at a coarse spatial (10km grain from IUCN basin-scale data) and temporal (30 year average) resolutions. Although the authors added several points in the discussion regarding these issues, I feel that some potential consequences on their estimates are too lightly addressed. For instance, I still don't think that the comparison with physiological estimates constitutes a 'Validation procedure' of their water temperature threshold. First, only about 3% of the fish species were considered. Second, although the authors did not find a directional bias (consistent underestimation or overestimation), the differences presented in Fig. S1 are of about 5-10 °C for many species, which can have a nonnegligible influence on their results. Another important source of uncertainty is related to the fact that species distributions are estimated using IUCN range maps. Beyond the fact that these maps are primarily derived from expert opinion (and thus that species do not necessary occur everywhere within the estimated range), they are provided at the sub-basin scale and so the resolution of the occurrence data are in fine lower than the 5 arcminutes resolution of the hydrological model. In turn, because hydrologic gradients are mainly defined along the longitudinal (upstream-downstream) gradient whereas climatic gradients usually show a stronger latitudinal structuring, the degree of uncertainty is likely to be much higher for the streamflow than climatic thresholds. This is not the first study using IUCN range maps to infer species distributions so I don't want to suggest that doing so is wrong in any ways. I agree with the authors that this database represents the best available data on freshwater fish species distributions. However, given that these maps were used to project a binary response between species persistence versus extirpation in the future, the degree of uncertainty around these thresholds can be consequential, and should not be overlooked.

Phylogenetic regression

I had difficulties following the different steps of the phylogenetic regression. First, I didn't understand why an average value of lambda was used and how it was obtained (across different models using one covariate at a time or using the full model but different response variables?). Perhaps references to previous work are needed here (beyond the reference to BIOMOD which I don't think is appropriate in this context). More generally, I suggest (again) to be more careful with the wording as this approach is purely correlative and thus does not allow to identify influential traits (e.g. L68) or explain/predict range changes (e.g. L126).

Reviewer #2:

Remarks to the Author:

Review of NCOMMS-19-17479A

This manuscript is a revised version of the manuscript Barbarossa et al. originally submitted. Going through the new manuscript and the response-to-reviewers document, it is clear that the authors have carefully considered our concerns and comments, resulting in an improved manuscript. The key improvements are: (1) considering a maximal dispersal scenario where each species is allowed to occupy all suitable grid cells within the watershed it is originally native to; (2) using phylogenetically informed statistical models to relate range size changes to species traits.

The authors also clearly state the limitations of their analysis, namely, underestimation of temperature and flow impacts of low order headwater streams, and ignoring potential impacts of flow seasonality or increased flows. I think the second issue is more important since the authors have already used the highest resolution data that is currently available. Impacts to fish native to arid regions due increased flows or reduced seasonality of flows may not be reflected in this analysis as stated by the authors.

While the addition of the phylogenetic models was a key improvement in the paper, I wasn't able to fully evaluate the models because information on the modeling procedure was inadequately presented. For example, I didn't understand the need to calibrate lambda values, and use the average value in

the final model, and it also wasn't clear what the final model was, and if there was any model selection procedure used in defining what constitutes the final model. Clarification on these methodological details is required.

The maximum dispersal scenario is an extreme scenario that is worth considering but it is unclear how realistic it is. The authors used an example of fish in the Rio Negro (a tributary of the Amazon) being able to disperse throughout the Amazon basin. The stream network in Amazon basin spans great distances and many fish species would likely not be able to disperse to all pixels within the basin. Incorporating dispersal to define which pixels are both suitable and reachable would probably be outside the scope of the paper given the amount of analyses and results already presented here. But would it be possible to use smaller watershed units, e.g., fish present in the Rio Negro can disperse within the Rio Negro or Rio Negro plus neighboring basins but not throughout the Amazon? In addition, I just want to double-check: regarding results of the maximum dispersal scenario analyses in Figs. 1-4, the changes in range size and changes in species numbers per grid cell are all expressed in losses and they are all capped at 0. Are there no grid cells with an increase in species, or no species that are projected to increase in range size?

I am signing my review for transparency.

-Xingli Giam

Specific comments

L44. Change "for" to "under". I also suggest change "...freshwater fish species worldwide (n=6,294)..." to "6294 freshwater fish species worldwide" to avoid confusion as to whether 6,294 represents all fish species globally. I also suggest a sentence after this to clarify that 6,294 is a subset (nevertheless a substantial one) of the 13,000-15000+ species [see Leveque et al. 2007 Global diversity of fish (Pisces) in freshwater. In Balian et al. (eds) Freshwater Animal Diversity Assessment pp 545-567 and also <https://www.iucnffsg.org/freshwater-fishes/freshwater-fish-diversity/> for estimates] globally with available data for the analysis.

L55. I checked ref. 21 and I do not think it appropriately supports the statement here. Ref. 21 is a study that aims to compare projected stream temperatures under climate change with fish thermal tolerances. It uses maximum weekly average temperatures but not the other two habitat variables used by this study (number of zero flow weeks and minimum weekly flow). The authors need to add a second citation of a study that links flow to species occupancy/survival.

L59. Thank you for clarifying your definition of mean standard error. To avoid confusion with the commonly used definition of standard error, I suggest naming this mean percent difference (multiplying this value by 100%). I also suggest adding the Pearson correlation coefficient value between max weekly water temperature and critical thermal maxima.

L77 and elsewhere. Suggest using "range size changes" rather than "range changes".

L82. I suggest changing "hotspots of area loss" to "hotspots of range losses".

Figure 1. Even with maximal dispersal, was the plotting of range losses capped at 0%? I would have expected at least some species to increase its range size given the extreme scenario that the fish would be able to expand its range to all cells with suitable temp and flow characteristics within the watershed.

Figure 2 caption. I suggest changing "area losses" to "range losses". Change "losing that cell" to "losing their ranges at that cell". Change "no data areas" to "areas with no available data".

L111. It is unclear whether the phylogenetic regression models are single variable models, or whether all predictor variables are included in a single model? (also see comment below on the phylogenetic regression method).

L132. Add "(where models predicted range changes instead of range losses under the no dispersal scenario)" to clarify what the response variable (because range changes are not constrained to losses under full dispersal).

L157. What is meant by "local flow regime"? Perhaps changing the sentence to "...adapted to low flows and specific seasonal flow regimes" might be clearer.

L184. Delete "a".

L224. I suggest replacing "employed" with "analysed".

L303. >50% of their current range? Or >50% of their current range *size*? I argue that we should calculate based on range *size* so that both scenarios can be compared.

L305. Similar to above, are the authors quantify the loss of ranges or changes in range size? I argue that with the addition of the max. dispersal scenario, calculating the latter would be more appropriate.

L305. When mapping species range losses at the grid cell level, what about grid cells that gain species range under the maximal dispersal scenario? A presentation and discussion of these results are missing.

L307-308. Should this value be named "change in species richness" rather than "area losses"? I think the former is a better description of the metric.

L328. There must be species present in both lotic and lentic systems. How did the authors classify these species?

L332. Why not use the raw trophic level values? Omnivorous fishes often exhibit really similar values to insectivorous fishes because the latter aren't feeding high up the food chain.

L338-342. The authors need to provide more details on the methods used to run the phylogenetic regression. I do not understand what the authors meant by "calibrating the lambda coefficient". Typically, Pagel's lambda (in the Pagel lambda phylogenetic model; there are other models that use different phylogenetic covariance structures) is estimated along with the covariates. Why did the authors use average lambda across different response variables? Overall, I didn't really understand how the phylogenetic models were run? I visited the GitHub account provided under the Data Availability section but I couldn't find the R code.

L342-344. What is the "final regression model"? Was a model selection or variable selection procedure performed before getting the final model? I do not think the computation of variable importance is correct. (1) The variable importance (if based on R^2) should be measured by R^2 of full model minus R^2 of model with the randomized variable (or simply with the variable removed) as this value reflects the drop of R^2 attributed to the loss of information from the model (Giam & Olden 2015 Ecological Modelling 313:307-313); (2) that said, the suitability of such a method is unclear because each predictor is probably related to the phylogeny as well; perhaps the $|t|$ -statistic of the phylogenetic regression might be an alternative measure of effect size (Giam & Olden 2017 Ecography 41:331-344).

Supplementary Methods. Details on the GCM-RCP scenarios. More details are needed. What are the 5 GCMs used? Supplementary Table 1 lists the abbreviations but will be good to list the full names here.

N.B. Line numbers refer to the clean version of the manuscript (not the one with highlighted changes)

Review comments

Handling editor

e1 Reviewer 1 remarks that inferring habitat suitability based only on maximum weekly water temperature, minimum weekly flow and the number of zero flow weeks is inappropriate; rather, they think that a better approach would have been to run species distribution models, which (while having their own limitations) better account for the multidimensional nature of species niches. After internal discussion, the other editors and I decided that we are willing to partly overrule this point, that is, we do not ask that you discard the current approach and start from scratch using species distribution models instead. However, we ask that you take the reviewer's concern as far as possible within the scope of this work, for example including additional hydrological variables (perhaps other flow metrics?) and interactions, and performing model selection. At the very least, a better justification of the particular variables used should be provided.

While we agree that species distribution models provide a more comprehensive representation of habitat suitability, we opted for our threshold-based mapping approach as we were interested in species' exposure to future climate extremes. We now better explain this in the paper (L. 44-47) and highly appreciate your willingness to maintain this approach. We further followed your advice by including two additional hydro-thermal extreme variables (minimum water temperature and maximum flow) and providing a better motivation for our selection of variables (Methods, L. 234-246).

e2) Both reviewers find the maximum dispersal scenario too unrealistic. We ask you address this by including also a more moderate dispersal scenario; perhaps some sensitivity analysis could also be helpful.

We added a more realistic dispersal scenario by defining dispersal boundaries based on the intersection of main hydrological basins and freshwater ecoregions, which account for evolutionary history and other ecological factors relevant to freshwater fishes (see further our reply to comments 1.2 and 2.3).

e3) Some of the methods were unclear to the referees, for instance they both had difficulties following all the steps of the phylogenetic regression models added in the revision. Please note that the Methods section does *not* contribute to the word limit, and it can therefore be expanded if that helps addressing the referees' concerns.

To address this concern, we extended and clarified the description of the phylogenetic regression in the methods section (see further our reply to comments 1.4 and 2.2).

Reviewer #1

This is the second time that I review the study from Barbarossa et al. and I have to say that it seems to be an entirely different manuscript. The presentation and the content are much improved, and the manuscript reads very well. I also appreciate the detailed responses to my previous comments and the fact that several limits of the study are now explicitly discussed in the main text. I think that the strength of the present study is to provide future water temperature and flow projections under a variety of scenarios; data that are so much needed (but too often lacking) to assess the vulnerability of freshwater organisms to climate change.

Thank you for these kind words.

However, I still have major reserves regarding the study design and the ecological interpretation of the results. The ecological metrics are very simplistic and some of the underlying assumption ecologically questionable. Although this study undoubtedly represents a useful starting point for future investigations, I am worried that the authors are trying to say too much based on their results, without taking the time to carefully consider the underlying assumptions and limits of their approach.

1.1) Terminology

First of all, I think that there is a confusion regarding the methodology that was used to estimate future range changes. The approach taken is fairly simplistic by considering that areas projected to be above/below the maximum or minimum water temperature and streamflow currently observed across species ranges will be unsuitable for the species in the future, resulting in range contractions. Conversely, the future expansion scenarios consider that species can expand in any grid cell present in the basin if the conditions stay below the previously identified thresholds. Given the approach I think that the terminology used in the manuscript (e.g. 'Modelling species-specific geographical range changes' (L278), 'we employed two dispersal assumption to model geographical range changes' (L290), 'we projected future range changes' L298, 'our model projections' L211) is misleading as it suggests that some sort of bioclimatic modelling approaches were used. I also question the comparison with the study of Warren et al. 2018 for terrestrial vertebrates (L205-207; also L302) as the later study did use modelling approaches (i.e. Maxent) to estimate future extirpation risk. Still related to this issue, I do not think that this study provides a 'comprehensive quantification of potential geographical range changes of freshwater fish in relation to future climate change' as claimed L135. In my opinion this study is not comprehensive either in terms of taxonomic coverage (~ 7000 species) or future climate change effects on geographical range (only maximum water temperature and minimum streamflow were considered). I thus suggest to lower the tone and reconsider the terminology used throughout the manuscript.

We agree with the issues raised and have made the following adjustments:

- We now refer to threats imposed by amplified extremes in water temperature and flow rather than geographic range changes, to avoid confusion with regards to the underlying methodology. We changed the terminology throughout the text (see e.g., L. 45-48) and the title of the manuscript.
- We removed the comparison of our results to those of Warren et al. 2018 as indeed their approach differed from ours.
- We took advantage of recently released species geographic range datasets to increase the number of species covered in our study. We assembled species ranges from a combination of the updated IUCN ranges, freshwater fish geographic ranges from multisource-compiled point records

(Barbarossa et al., 2020) and the recently released Amazonfish dataset (Jézéquel et al., 2020). Our dataset now comprises about 11,500 lotic fish species (~90% of the known number of freshwater fish species). See Methods section “Species occurrence data” (L. 192-210) for further details.

- We have extended our set of climate extreme variables by adding minimum water temperature and maximum flow. In addition, we quantified hydro-thermal niche variables representing seasonality (as key factors of habitat suitability), which we did not include in the modelling because we found correlations of 0.6-0.9 (Pearson’s r) between these variables and our minimum flow and water temperature limits (see below Figure R1 and L. 146-148).

Figure R1. Correlation matrix (Pearson’s r) across species-specific thresholds. In the figure above, Q is flow and T water temperature, min, max and cv stand for minimum, maximum, and coefficient of variation, respectively, while Qzf stands for number of zero flow weeks. For coefficients of variation both the upper (up) and lower (low) thresholds are considered. The thresholds of the other variables are established according to the description in the Methods section. It should be noted that the high correlation between Qzf (number of zero flow weeks) and Qmin is expected as these two metrics are complementary. Indeed, when minimum annual weekly flow is 0 (seasonal behavior of the stream), then Qzf > 0 and vice-versa.

1.2) Ecological assumptions

The ecological assumptions behind the approach are also questionable as they assume that maximum weekly water temperature, minimum weekly flow and the number of zero flow weeks are the only factors affecting the distribution of freshwater fishes and that they act independently from each other. Although the approach can be defensible regarding future extirpation risk (though many caveats must be acknowledged as I highlighted in my previous round of comments), I believe that the resulting assumptions for the dispersal scenarios are ecologically untenable. For instance, species can disperse to any given pixel as defined by the maximum weekly temperature predicted in the future

with no consideration whatsoever for hydrological or other habitat conditions. I understand that the authors included an estimation of future range expansion in an attempt to respond to the concerns raised by the reviewer #2, but I don't think that the approach taken is the right answer. If the goal of the study is to assess potential changes in habitat suitability in the future, then the authors may want to consider using species distribution models. I do not want to imply here that these models are perfect tools but they have at least the merit to account for the multidimensional nature of species niches.

To address these concerns we made the following adjustments:

- We added two more climate extreme variables (see also our reply to 1.1) and we explain better why we focus on extremes, i.e., because these are indeed more indicative of extirpation risk than other climate characteristics (L. 45-47).
- In the discussion, we better highlight the assumptions that need to be met for exposure to amplified extremes to result in extirpation, as follows (L. 149-158): "Exposure to climate extremes beyond present-day values does not necessarily imply local extinction. If species' current distributions are confined by factors other than flow or water temperature (e.g., biogeographic dispersal barriers or anthropogenic pressures), species might be able to withstand larger temperature and flow extremes than inferred based on their current geographic range^{25,26}. The same holds if species are able to adapt to new water temperature and flow conditions¹⁶ or if fishes have the possibility to hide from extremes in micro-climatic refugia, for example due to water stratification or small-scale thermal heterogeneity²⁷, which are not included in our hydrological model²⁰. On the other hand, species' range maps are relatively coarse representations of species occurrence, hence some species might be more affected than indicated by our results (i.e., if present-day climate extremes within their geographic range already preclude local occurrence)."
- We modified the "maximal dispersal" scenario to better reflect other factors limiting the distribution of freshwater fishes. To that end, we now define dispersal boundaries using a combination of main basin boundaries (hard physical boundary) and freshwater ecoregions of the world (representing additional evolutionary and ecological factors delimiting the distribution of freshwater fish species) (Figure S12 and L. 300-306), instead of main basin boundaries only.

1.3) Estimation of habitat suitability thresholds

I also still have reserves regarding the estimation of habitat suitability thresholds. These were derived using empirical data defined at a coarse spatial (10km grain from IUCN basin-scale data) and temporal (30 year average) resolutions. Although the authors added several points in the discussion regarding these issues, I feel that some potential consequences on their estimates are too lightly addressed. For instance, I still don't think that the comparison with physiological estimates constitutes a 'Validation procedure' of their water temperature threshold. First, only about 3% of the fish species were considered. Second, although the authors did not find a directional bias (consistent underestimation or overestimation), the differences presented in Fig. S1 are of about 5-10 °C for many species, which can have a nonnegligible influence on their results. Another important source of uncertainty is related to the fact that species distributions are estimated using IUCN range maps. Beyond the fact that these maps are primarily derived from expert opinion (and thus that species do not necessary occur everywhere within the estimated range), they are provided at the sub-basin scale and so the resolution of the occurrence data are in fine lower than the 5 arcminutes resolution of the hydrological model. In turn, because hydrologic gradients are mainly defined along the longitudinal (upstream-downstream) gradient whereas climatic gradients usually show a stronger latitudinal structuring, the degree of uncertainty is likely to be much higher for the

streamflow than climatic thresholds. This is not the first study using IUCN range maps to infer species distributions so I don't want to suggest that doing so is wrong in any ways. I agree with the authors that this database represents the best available data on freshwater fish species distributions. However, given that these maps were used to project a binary response between species persistence versus extirpation in the future, the degree of uncertainty around these thresholds can be consequential, and should not be overlooked.

We understand these two concerns (i.e., limited validation data and relatively large differences between our thresholds and the lab data) and made the following adjustments:

- We have searched additional data on thermal tolerance values from lab experiments, which enabled us to nearly double the number of species represented in the comparison (Figure S8).
- In the discussion we now elaborate more on the deviations between our thresholds and the lab data, as follows (L. 158-165): " Indeed, a tentative comparison of our species-specific maximum weekly water temperature limits with critical thermal maxima reported from laboratory tests suggest both under- and overestimations by our geographic range-based thermal limits, while showing an overall reasonable agreement (mean percent difference = 9%; Pearson's $r = 0.62$; Figure S8). Further work is required to better understand deviations between our empirical thresholds and the lab-based maxima, which may stem not only from uncertainties in our modelling approach (e.g., in the range maps or the water temperature model) but also from heterogeneity in experimental conditions²⁸."
- In addition, in the discussion we now better acknowledge the implications of uncertainties in our range-based approach for the potential to assess extirpation (L. 149-158; see also our reply to comment 1.2).

1.4) Phylogenetic regression

I had difficulties following the different steps of the phylogenetic regression. First, I didn't understand why an average value of lambda was used and how it was obtained (across different models using one covariate at a time or using the full model but different response variables?). Perhaps references to previous work are needed here (beyond the reference to BIOMOD which I don't think is appropriate in this context). More generally, I suggest (again) to be more careful with the wording as this approach is purely correlative and thus does not allow to identify influential traits (e.g. L68) or explain/predict range changes (e.g. L126).

Thank you for pointing at these unclear passages in our methods section. We have extended the description of the different steps carried out in the multiple phylogenetic regression and added more references, at L. 356-369: "We ran a multiple regression between the threat level (response variable) and the species characteristics (covariates) for different warming levels and dispersal scenarios (i.e., one multiple regression model per warming level-dispersal combination, eight in total). Prior to running the regressions we log-transformed range area and body length as these variables were highly skewed. As Spearman's rank correlations among the covariates were below 0.4 and variance inflation factors below 1.5 (Figure S13 and Table S6), we kept the full set. We ran the phylogenetic regression using the R package "nlme"^{60,61} and extracted coefficients, $|t|$ -statistics, p-values as well as the lambda parameter at each warming level (Table S5). Then, we quantified variable importance using a procedure based on the random forest approach⁶², as implemented in the R package biomod2⁶³. To that end, we randomized the values of the covariates one by one and computed the variable importance as 1 minus the Pearson's r between the predictions of the original model and the predictions obtained from the model with randomized data. We iterated this procedure 100 times for each variable and each of the eight models (four warming levels times two dispersal assumptions)

and reported the average score (standard deviation across the iterations was negligible).” In addition, we changed the wording of ‘influential traits’ to ‘relationships with traits’ throughout, to be more explicit on the correlative nature of the analysis (e.g., L. 109-111).

Reviewer #2

This manuscript is a revised version of the manuscript Barbarossa et al. originally submitted. Going through the new manuscript and the response-to-reviewers document, it is clear that the authors have carefully considered our concerns and comments, resulting in an improved manuscript. The key improvements are: (1) considering a maximal dispersal scenario where each species is allowed to occupy all suitable grid cells within the watershed it is originally native to; (2) using phylogenetically informed statistical models to relate range size changes to species traits.

2.1) The authors also clearly state the limitations of their analysis, namely, underestimation of temperature and flow impacts of low order headwater streams, and ignoring potential impacts of flow seasonality or increased flows. I think the second issue is more important since the authors have already used the highest resolution data that is currently available. Impacts to fish native to arid regions due increased flows or reduced seasonality of flows may not be reflected in this analysis as stated by the authors.

Thanks for raising this issue. We have extended our set of variables representing climate extremes to also include maximum flow (L. 244-246). In addition, we quantified hydro-thermal niche variables representing seasonality (as key factors of habitat suitability) and found correlations of 0.6-0.9 (Pearson's r) between these variables and our minimum flow and temperature limits and therefore decided to exclude them from our analysis (see Figure R1 above and L. 146-148) (see also our reply to comment 1.1).

2.2) While the addition of the phylogenetic models was a key improvement in the paper, I wasn't able to fully evaluate the models because information on the modeling procedure was inadequately presented. For example, I didn't understand the need to calibrate lambda values, and use the average value in the final model, and it also wasn't clear what the final model was, and if there was any model selection procedure used in defining what constitutes the final model. Clarification on these methodological details is required.

Thank you for raising these issues. Lambda values are now estimated for each warming level-dispersal scenario combination separately (see also our reply to 2.22) and are also reported in Figure 5 and Table S5. We now also better explain that we run a multiple regression model based on a set of potentially relevant covariates of which we checked potential multicollinearity upfront. We agree that the term 'final model' is confusing and now consistently refer to 'multiple regression model'. We have rewritten the methods section as follows (L. 330-369): "We performed a multiple linear regression analysis to relate the threat level of each species, quantified as the proportion of the geographic range exposed to future climate extremes beyond current levels within the range, to a number of potentially relevant species characteristics. We used phylogenetic regression to account for non-independence of observations due to phylogenetic relatedness among species⁵⁵. Information on phylogeny was available for a subset of 4,930 fish species employed in our study from Betancur-R et al.⁵⁶, and therefore the phylogenetic regression analysis was performed on this sample. As species characteristics, we included initial range size (in km²), body length (in cm), climate zone, trophic group and habitat type, as these traits may influence species' responses to (anthropogenic) environmental change^{8,28,57,58}. We further included IUCN Red List category to evaluate the extent to which current threat status is indicative of potential impacts of future climate change, and commercial importance to evaluate implications of potential extirpations for fisheries. We overlaid each species' occurrence range with the historic Köppen-Geiger climate categories to obtain the main climate zone per species (i.e., capital letter of the climate classification)⁵⁹. Species falling into

multiple climate categories were assigned the climate zone with the largest overlap. We retrieved information on threat status from IUCN³⁸ and on taxonomy from Fishbase⁴¹. We used the IUCN and Fishbase data also to gather a list of potential habitats for each species. For the species represented within the IUCN dataset, we classified species as lotic if they were associated with habitats containing at least one of the words “river”, “stream”, “creek”, “canal”, “channel”, “delta”, “estuaries”, and as lentic if the habitat descriptions contained at least one of the words “lake”, “pool”, “bog”, “swamp”, “pond”. For the remaining species, we extracted information on habitat from Fishbase, where we classified species found in lakes as lentic and species found in rivers as lotic. We classified species occurring in both streams and lakes as lotic-lentic and labeled species found in both freshwater and marine environments as lotic-marine. Further, we retrieved data from Fishbase on maximum body length and commercial importance⁴¹. From the same database we also retrieved trophic level values and aggregated them into Carnivore (trophic level >2.79), Omnivore (2.19 < trophic level ≤ 2.79) and Herbivore (trophic level ≤ 2.19)⁴¹. We performed a synonym check for the binomial nomenclature provided by the IUCN database to maximize the overlap with the Fishbase database. We ran a multiple regression between the threat level (response variable) and the species characteristics (covariates) for different warming levels and dispersal scenarios (i.e., one multiple regression model per warming level-dispersal combination, eight in total). Prior to running the regressions we log-transformed range area and body length as these variables were highly skewed. As Spearman’s rank correlations among the covariates were below 0.4 and variance inflation factors below 1.5 (Figure S13 and Table S6), we kept the full set. We ran the phylogenetic regression using the R package “nlme”^{60,61} and extracted coefficients, |t|-statistics, p-values as well as the lambda parameter at each warming level (Table S5). Then, we quantified variable importance using a procedure based on the random forest approach⁶², as implemented in the R package biomod2⁶³. To that end, we randomized the values of the covariates one by one and computed the variable importance as 1 minus the Pearson’s r between the predictions of the original model and the predictions obtained from the model with randomized data. We iterated this procedure 100 times for each variable and each of the eight models (four warming levels times two dispersal assumptions) and reported the average score (standard deviation across the iterations was negligible).”

2.3) The maximum dispersal scenario is an extreme scenario that is worth considering but it is unclear how realistic it is. The authors used an example of fish in the Rio Negro (a tributary of the Amazon) being able to disperse throughout the Amazon basin. The stream network in Amazon basin spans great distances and many fish species would likely not be able to disperse to all pixels with the basin. Incorporating dispersal to define which pixels are both suitable and reachable would probably be outside the scope of the paper given the amount of analyses and results already presented here. But would it be possible to use smaller watershed units, e.g., fish present in the Rio Negro can disperse within the Rio Negro or Rio Negro plus neighboring basins but not throughout the Amazon? In addition, I just want to double-check: regarding results of the maximum dispersal scenario analyses in Figs. 1-4, the changes in range size and changes in species numbers per grid cell are all expressed in losses and they are all capped at 0. Are there no grid cells with an increase in species, or no species that are projected to increase in range size?

Following your suggestion, we revised the dispersal boundaries in our maximal dispersal scenario. To that end, we combined the boundaries of the main hydrological basins (e.g., the watershed delimiting the Amazon basin) with those of the freshwater ecoregions of the world (FEOW), enabling us to break down larger basins like the Amazon in smaller sub-units based on similarity in evolutionary history and other ecological factors relevant to freshwater fishes. A map of the resulting

sub-basin division is provided in Figure S12. A more detailed description of the procedure used to delineate the sub-basin units is provided in the methods section “Accounting for dispersal”.

In the dispersal scenario, we do not cap the values to zero. We first expand ranges to the full extent of the underlying sub-basin units and then calculate the proportion of the range threatened due to alterations in extreme hydro-thermal conditions. This is now better explained in the methods section (L. 306-311). This means that inter-scenario gains in range size (incl. relative to present) are possible, e.g., species X shows reduction in range size of 5% at 1.5°C and then 2% at 2°C resulting in a 3% gain at 2°C). Overall, our results show that all species will have at least a small portion of range size affected by the alteration in hydro-thermal extremes and therefore inter-scenario gains are hardly visible from the presented charts, but they are present.

I am signing my review for transparency.

-Xingli Giam

Specific comments

2.4) L44. Change “for” to “under”. I also suggest change “...freshwater fish species worldwide (n=6,294)...” to “6294 freshwater fish species worldwide” to avoid confusion as to whether 6,924 represents all fish species globally. I also suggest a sentence after this to clarify that 6,294 is a subset (nevertheless a substantial one) of the 13,000-15000+ species [see Leveque et al. 2007 Global diversity of fish (Pisces) in freshwater. In Balian et al. (eds) Freshwater Animal Diversity Assessment pp 545-567 and also <https://www.iucnffsg.org/freshwater-fishes/freshwater-fish-diversity/> for estimates] globally with available data for the analysis.

Thanks for your comment; we have adjusted the wording (L. 44-45): “Here, we assess future climate threats to 11,425 riverine fish species by quantifying their exposure to flow and water temperature extremes under different global warming scenarios.” In addition, as you can see from the new version of the manuscript, we have compiled a novel global geographic ranges dataset for ~13,000 freshwater fish species, now covering ~90% of the known freshwater fish species (see section “Species occurrence data” in the Methods and reply to 1.1 for details).

2.5) L55. I checked ref. 21 and I do not think it appropriately supports the statement here. Ref. 21 is a study that aims to compare projected stream temperatures under climate change with fish thermal tolerances. It uses maximum weekly average temperatures but not the other two habitat variables used by this study (number of zero flow weeks and minimum weekly flow). The authors need to add a second citation of a study that links flow to species occupancy/survival.

We now provide a more detailed description of the rationale behind the streamflow and temperature variables employed in our study, with more specific references, at L. 234-246.

2.6) L59. Thank you for clarifying your definition of mean standard error. To avoid confusion with the commonly used definition of standard error, I suggest naming this mean percent difference (multiplying this value by 100%). I also suggest adding the Pearson correlation coefficient value between max weekly water temperature and critical thermal maxima.

We adjusted the metric name accordingly and added the Pearson's r .

2.7) L77 and elsewhere. Suggest using “range size changes” rather than “range changes”.

We changed the general framing of the paper from “range changes” to “proportion of the range threatened by future climate extremes” (see also our reply to 1.1).

2.8) L82. I suggest changing “hotspots of area loss” to “hotspots of range losses”.

Following the change in framing, we now refer to hotspots of future climate threat.

2.9) Figure 1. Even with maximal dispersal, was the plotting of range losses capped at 0%? I would have expected at least some species to increase its range size given the extreme scenario that the fish would be able to expand its range to all cells with suitable temp and flow characteristics within the watershed.

Our approach does account for potential range expansions, i.e., values have not been capped (see our reply to 2.3).

2.10) Figure 2 caption. I suggest changing “area losses” to “range losses”. Change “losing that cell” to “losing their ranges at that cell”. Change “no data areas” to “areas with no available data”.

We changed the caption in accordance with our new framing. The caption now reads: “Potentially affected fraction (PAF) of freshwater fish species due to exposure to water flow and temperature extremes beyond current levels, for different global warming levels and two dispersal assumptions. Patterns are based on the median PAF across the GCM-RCP combinations at a five arcminutes resolution (~10 km at the Equator). Gray denotes no data areas (no species occurring or no data available). Source data are provided as a Source Data file.”

2.11) L111. It is unclear whether the phylogenetic regression models are single variable models, or whether all predictor variables are included in a single model? (also see comment below on the phylogenetic regression method).

Thank you for raising this issue. We now specify at L. 109 that we use multiple phylogenetic regression, so one model where all covariates are included. In addition, in the methods we specify that (L. 356-359) “We ran a multiple regression between the threat level (response variable) and the species characteristics (covariates) for different warming levels and dispersal scenarios (i.e., one multiple regression model per warming level-dispersal combination, eight in total).”

2.12) L132. Add “(where models predicted range changes instead of range losses under the no dispersal scenario)” to clarify what the response variable (because range changes are not constrained to losses under full dispersal).

This sentence has been removed in the revision.

2.13) L157. What is meant by “local flow regime”? Perhaps changing the sentence to “...adapted to low flows and specific seasonal flow regimes” might be clearer.

We changed this to (L. 145-146): “(...) which might disfavor species whose life histories are adapted to specific flow or temperature regimes (e.g., specific seasonal flow regimes)^{5,24}.”

2.14) L184. Delete “a”.

Adjusted.

2.15) L224. I suggest replacing “employed” with “analysed”.

Adjusted.

2.16) L303. >50% of their current range? Or >50% of their current range *size*? I argue that we should calculate based on range *size* so that both scenarios can be compared.

Thank you for pointing this out. Results are indeed based on range size. However, we now refer to the proportion of (expanded) range threatened, in accordance with the adjusted framing (e.g., L. 313-315).

2.17) L305. Similar to above, are the authors quantify the loss of ranges or changes in range size? I argue that with the addition of the max. dispersal scenario, calculating the latter would be more appropriate.

We now refer to proportion of (expanded) range threatened by future climate extremes.

2.18) L305. When mapping species range losses at the grid cell level, what about grid cells that gain species range under the maximal dispersal scenario? A presentation and discussion of these results are missing.

See our response to 2.3 and 2.9.

2.19) L307-308. Should this value be named “change in species richness” rather than “area losses”? I think the former is a better description of the metric.

We agree that the terminology used for the metric was confusing. We now call it potentially affected fraction (L. 317-328).

2.20) L328. There must be species present in both lotic and lentic systems. How did the authors classify these species?

Indeed there are species occurring in both systems (which we now label as lotic-lentic species). We now better explain the species classification in the methods section at L. 344-352: “We used the IUCN and Fishbase data also to gather a list of potential habitats for each species. For the species represented within the IUCN dataset, we classified species as lotic if they were associated with habitats containing at least one of the words “river”, “stream”, “creek”, “canal”, “channel”, “delta”, “estuaries”, and as lentic if the habitat descriptions contained at least one of the words “lake”, “pool”, “bog”, “swamp”, “pond”. For the remaining species, we extracted information on habitat from Fishbase, where we classified species found in lakes as lentic and species found in rivers as lotic. We classified species occurring in both streams and lakes as lotic-lentic and labeled species found in both freshwater and marine environments as lotic-marine.”

2.21) L332. Why not use the raw trophic level values? Omnivorous fishes often exhibit really similar values to insectivorous fishes because the latter aren’t feeding high up the food chain.

The records of trophic levels in Fishbase are quite incomplete (62% of NAs for the subset of 4,930 species used in the phylogenetic regression). By categorizing the variable, we can assign a data deficient (DD) category to the NA records and then still use the variable in the phylogenetic

regression. So, the reason is practical (data availability) rather than ecological. The alternative would be to exclude this variable.

2.22) L338-342. The authors need to provide more details on the methods used to run the phylogenetic regression. I do not understand what the authors meant by “calibrating the lambda coefficient”. Typically, Pagel’s lambda (in the Pagel lambda phylogenetic model; there are other models that use different phylogenetic covariance structures) is estimated along with the covariates. Why did the authors use average lambda across different response variables? Overall, I didn’t really understand how the phylogenetic models were run? I visited the GitHub account provided under the Data Availability section but I couldn’t find the R code.

Thank you for pointing this out. Given the large number of observations and consequential high-dimensionality of the covariance matrix, we had opted for calibrating the lambda coefficient on one warming scenario (there are eight scenarios in total, four with and four without dispersal) and extrapolate that value to the others to spare computing time (lambda was about constant around the value of 0.95 after some manual testing). Following your comment, we now estimate lambda along with the model parameters for each of the eight multiple regression models separately. We clarified the overall procedure in the methods at L. 356-369 (see reply to 2.2). The scripts concerning the phylogenetic regression are available at the GitHub page <https://github.com/vbarbarossa/fishsuit> (I made sure it is now fully visible and updated), at scripts/R/res/phyloreg.R.

2.23) L342-344. What is the “final regression model”? Was a model selection or variable selection procedure performed before getting the final model? I do not think the computation of variable importance is correct. (1) The variable importance (if based on R^2) should be measured by R^2 of full model minus R^2 of model with the randomized variable (or simply with the variable removed) as this value reflects the drop of R^2 attributed to the loss of information from the model (Giam & Olden 2015 Ecological Modelling 313:307-313) ; (2) that said, the suitability of such a method is unclear because each predictor is probably related to the phylogeny as well; perhaps the $|t|$ -statistic of the phylogenetic regression might be an alternative measure of effect size (Giam & Olden 2017 Ecography 41:331-344).

We adjusted the text to clarify these steps (see also our reply to 2.2). We ran multiple regressions with all the species characteristics as independent variables, whereby we decided not to exclude any variables upfront because multicollinearity was rather low (VIFs < 1.5, Table S6). We agree that the term ‘final model’ is confusing and now refer to ‘multiple regression model’ instead. We now also better explain our method to quantify variable importance, which is based on Random Forests (see L. 363-369): “Then, we quantified variable importance using a procedure based on the random forest approach⁶², as implemented in the R package biomod2⁶³. To that end, we randomized the values of the covariates one by one and computed the variable importance as 1 minus the Pearson’s r between the predictions of the original model and the predictions obtained from the model with randomized data. We iterated this procedure 100 times for each variable and each of the eight models (four warming levels times two dispersal assumptions) and reported the average score (standard deviation across the iterations was negligible).” In addition, we also extracted the $|t|$ -statistics from the model output, as suggested (Table S5).

2.24) Supplementary Methods. Details on the GCM-RCP scenarios. More details are needed. What are the 5 GCMs used? Table S1 lists the abbreviations but will be good to list the full names here.

This part of the supplementary methods now describes which GCMs were used and provides more detail on the ISI-MIP project, which tailors GCM output for use in impact models (such as PCR-GLOBWB).

Reviewers' Comments:

Reviewer #1:

Remarks to the Author:

I am satisfied with the revisions made by the authors, which address my previous concerns. I am not particularly an advocate of species distribution models but I think that changing the terminology is now more in line with what has been done. Providing more justifications for their approach also results in an improved manuscript. I also very much appreciate their efforts to expand their dataset in terms of species distributions, climate variables, C_{max} and dispersal scenarios.

However, I do have a remaining concern regarding the results for the maximum dispersal scenario. Looking at Fig. 3, I noticed that the potentially affected fraction of species per hydrographic basin is predicted to increase from the no dispersal to maximum dispersal scenario in several basins (e.g. Mississippi, Orinoco, Uruguay, Niger, Congo, Yenisey). I spotted the same problem in Fig. S4. Unless I misunderstood something, I don't think that this should ever be expected (cases where more species are expected to be at risk when they show some degree of adaptive capacity than when they don't). Thinking about it, I realized that it might come from the way the 'threatened' cells are identified. Indeed, it is stated L306-311 that future climate threat under this scenario was estimated using the differences between the expanded range (species occupy all grid cells within the sub-watershed) and the future range where grid cells above the climatic thresholds are set to zero. This might be problematic if the grid cells that are currently unoccupied within the sub-watershed but predicted to be unsuitable in the future (above the thresholds) are counted in the estimate. This might result in an overestimation of the climate risk under this scenario. For instance, a species will be predicted to experience an increase in climate risk even if none of the currently occupied pixels are flagged as threatened in the future, whereas they should instead be predicted to experience an increase in geographical range size. I think my concern is in line with the previous comment of R#2 regarding the fact that the indices are all capped at 0. These pixels (currently unoccupied within the same sub-watershed but at risk in the future) should be removed from the estimates as species will likely not colonize an area that will be unsuitable. Perhaps this was already done and the problem comes from something else (?). This needs to be clarified and corrected.

My second comment is related to the taxonomic coverage of the phylogenetic multiple regressions. Out of the 11500 species included in the study, only about 5000 are included in the trait analysis, which is unfortunate given the efforts deployed to expand the geographic range dataset. The authors might not be aware of it, but a comprehensive phylogeny of fish is now available, which can be accessed using the R package fishtree (I'm sorry I didn't think about this resource during the previous rounds of review). This might be worth considering as I would expect that most species will be included in this phylogeny (after taxonomic harmonization).

Specific comments:

L90-93: I didn't understand these two sentences, although I have the feeling that this is related to my previous comment regarding the maximum dispersal scenario (?).

L146-148: Perhaps it might be worth developing this point further (briefly).

L279 Change 'modelling' for 'projecting'?

L292 Wouldn't 'at risk' be a better choice than 'threatened'?

L299-311 Both 'sub-watershed' and 'sub-basin' are used in this paragraph. I suggest to use 'sub-basin' throughout the manuscript.

L319 I suggest to replace the subscript 's' by another letter to avoid confusion with the number of species 'S'

L348-350 This is a detail but is the information provided in fishbase only binary: 'lake' versus 'rivers'? I thought that it was more similar to what is described for the IUCN datasets

L356 I suggest to use the plural 'we ran multiple regressions'

Figure 4. As I understand it, only pixels flagged as threatened by both flow and temperature thresholds are shown. Perhaps showing the total overlap of the two threats in the bottom panel would be more informative as it is ultimately what has been used in the climate threat analysis? At least the legend should more clearly state what is represented by 'both'

Figure 5 – The reference should be to Table S5 not S3. I am also unsure of how the Pearson's r reported in the legend were obtained (same question Figure S7).

Figure S11- (b) should be the percentage instead of ratio?

Reviewer #2:

Remarks to the Author:

I have gone through the response-to-reviewers letter and the revised manuscript. The authors have incorporated my suggestions and addressed my concerns in their revisions. I think this paper would be an excellent contribution to the literature on the global impact of climate change on freshwater fish.

I have just one more concern/question now that the authors have clarified their phylogenetic modeling approach. My concern is about the response variable in the phylogenetic models. The response variable is a proportion (bounded between 0-1). Proportions are notoriously difficult to model without transformations, and because the authors didn't mention performing any model diagnostics/model assumption checks, I am wondering if the residuals might actually be non-normal, which would violate a key assumption of the model. I suggest the authors make a Q-Q plot of the residuals to check for residual normality along with other standard model diagnostics plots (e.g., residual vs. fitted plots to check for linearity and homogeneity in variance). If model residuals are indeed non-normal (e.g., highly skewed), a log-transform of proportion or using log-response-ratio [i.e., $\log(\text{range size threatened by climate change}/\text{range size not threatened by climate change})$] might help improve/ensure normality of residuals and be more appropriate as a response variable.

Other than the abovementioned concern, I only have a few more minor suggestions and recommendations to help make the paper clearer for readers.

-Xingli Giam

Specific comments (line numbers refer to the ones on the document where changes were tracked)

L24. Suggest changing to "In a world that is 3.2°C warmer (if current national targets for reductions in greenhouse gas emissions for 2030 under the Paris Agreement are met), ..." to give more context and background into pledges by governments under the Paris Agreement.

L85. Suggest clarifying that 3.2°C warming value is the warming limit by the end of the century with 66% confidence (UN Emissions Gap Report 2019, Pg XIX, last bullet point), i.e., there is a 66% chance that warming will be $\leq 3.2^\circ\text{C}$ by 2100 if current unconditional Nationally Determined Contributions (NDC) are fully implemented.

L85-86. Suggest changing to "...set at 3.2°C warming value. This value represents the maximum warming limit predicted to occur by the end of the century (with 66% probability) if all current greenhouse gas emissions reductions targets (unconditional Nationally Determined Contributions;

NDC) for 2030 are met (cite UN Emissions Gap Report 2019).

L133-134. "Under the maximal dispersal scenario...". This section lacks a comparison with the "no dispersal" scenario. Do the authors mean that the maximal and no dispersal scenarios show similar spatial patterns but with slightly lower threat levels distributed over larger areas? If so, then change to "Under the maximal dispersal scenario, threatened areas are similar to those under the no dispersal scenario but with lower threat levels distributed over larger areas than in the no dispersal scenario."

L149-156. The authors presented results in areas affected by changes in low flow. How about changes in high flow?

L157. Does "amplified flow" here mean both low and high flow? It might be confusing because on L144, the authors mentioned "amplified low or high flow conditions".

L179. I looked at Table S5-II and interestingly, the threat to climate change was positively associated with the (ordinal levels) of threat status. Coefficients were always $LC < NT < VU < EN < 0$ (CR ref. level) with the exception of 1.5°C warming scenario. Even then LC and NT coefficients were very similar. I wonder if this is worth mentioning in the results and the discussion.

Table S5-II. In the caption, I suggest including information with which factor level is the reference level by indicating for example, A=Equatorial (reference level); CR=Critically endangered (reference level), etc.

L202-203. Suggest changing "amplified high water temperatures" to "the increase in maximum water temperatures".

L204. Suggest replacing "This reflects that" with "This is because".

L226. Suggest insert "explicitly" between "did not" and "consider".

L230. The author can consider inserting at the end of the current sentence ", indicating that the effects of seasonal flow or temperature changes were likely at least partially incorporated in our predictions."

L253. Correct typo error: replace "treat" with "threat".

L547. From what I understand, the authors ran only a phylogenetic regression model, right? If that is the case, replace "multiple linear regression analysis" with "phylogenetic regression analysis". Combine the first and second sentences on this page: "We performed a phylogenetic regression analysis to relate...to a number of potentially relevant species characteristics while accounting for the non-independence of observations due to phylogenetic relatedness among species."

L548. The response variable is a proportion value (bounded between 0-1). This is potentially an issue because the model residuals are often far from normally distributed, which is an important assumption of the phylogenetic regression model. The authors did not mention checking of model assumptions and or performing standard model diagnostics. I suggest the authors make a Q-Q plot of the model residuals to check for normality of residuals and plot residuals against fitted values to check for linearity and homogeneity in variance. From experience, modeling proportions might be really tricky. If the residuals are not normally distributed, log-transforming the proportions might work in some cases but not all. If that doesn't work, using the log-response-ratio [i.e., $\log(\text{range size threatened by climate change}/\text{range size not threatened by climate change})$] might work better.

L573. Suggest replacing "multiple regression" with "individual phylogenetic regression models".

N.B. Line numbers refer to the clean version of the manuscript (not the one with highlighted changes)

Review comments

Reviewer #1

I am satisfied with the revisions made by the authors, which address my previous concerns. I am not particularly an advocate of species distribution models but I think that changing the terminology is now more in line with what has been done. Providing more justifications for their approach also results in an improved manuscript. I also very much appreciate their efforts to expand their dataset in terms of species distributions, climate variables, C_{max} and dispersal scenarios.

Thank you for the appreciation.

1.1) However, I do have a remaining concern regarding the results for the maximum dispersal scenario. Looking at Fig. 3, I noticed that the potentially affected fraction of species per hydrographic basin is predicted to increase from the no dispersal to maximum dispersal scenario in several basins (e.g. Mississippi, Orinoco, Uruguay, Niger, Congo, Yenisey). I spotted the same problem in Fig. S4. Unless I misunderstood something, I don't think that this should ever be expected (cases where more species are expected to be at risk when they show some degree of adaptive capacity than when they don't). Thinking about it, I realized that it might come from the way the 'threatened' cells are identified. Indeed, it is stated L306-311 that future climate threat under this scenario was estimated using the differences between the expanded range (species occupy all grid cells within the sub-watershed) and the future range where grid cells above the climatic thresholds are set to zero. This might be problematic if the grid cells that are currently unoccupied within the sub-watershed but predicted to be unsuitable in the future (above the thresholds) are counted in the estimate. This might result in an overestimation of the climate risk under this scenario. For instance, a species will be predicted to experience an increase in climate risk even if none of the currently occupied pixels are flagged as threatened in the future, whereas they should instead be predicted to experience an increase in geographical range size. I think my concern is in line with the previous comment of R#2 regarding the fact that the indices are all capped at 0. These pixels (currently unoccupied within the same sub-watershed but at risk in the future) should be removed from the estimates as species will likely not colonize an area that will be unsuitable. Perhaps this was already done and the problem comes from something else (?). This needs to be clarified and corrected.

We agree that areas outside the current range of the species that will be threatened in the future scenario for that species, should not count towards the total amount of cells threatened. We adjusted the equation, reran the simulations and updated figures and tables accordingly. We now describe in more detail how the percentage of range threatened is calculated for each of the two scenarios (lines 287-290; lines 318-322): "Thus, for each species x we quantified the percentage of geographic range threatened (RT [%]) at each GCM-RCP scenario combination c and for a variable (or group of variables) v , as

$$RT_{x,c,v} = \frac{AT_{x,c,v}}{A_x} \cdot 100 \quad (\text{Eq. 6})$$

where AT is the area threatened [km^2] and A is the original geographic range [km^2]." (.....) "Then we assessed future climate threats for the 42 different scenarios relative to the present-day range plus all cells potentially available to the species within the encompassing sub-basins (excluding cells that would become threatened in the future), as

$$RT_{x,c,v} = \frac{AT_{x,c,v}}{A_x + (AE_x - AET_{x,c,v})} \cdot 100 \quad (\text{Eq. 7})$$

where AE is the expanded part of the geographic range [km²] and AET is the area threatened within the expanded part of the geographic range [km²].”

1.2) My second comment is related to the taxonomic coverage of the phylogenetic multiple regressions. Out of the 11500 species included in the study, only about 5000 are included in the trait analysis, which is unfortunate given the efforts deployed to expand the geographic range dataset. The authors might not be aware of it, but a comprehensive phylogeny of fish is now available, which can be accessed using the R package fishtree (I’m sorry I didn’t think about this resource during the previous rounds of review). This might be worth considering as I would expect that most species will be included in this phylogeny (after taxonomic harmonization).

Thank you for the suggestion. We have checked the R package fishtree and found that it is based on the same phylogeny from Betancur-R et al., 2017 as already used in our study, yet provides a stochastically-generated complete phylogeny. As suggested, we now use the phylogeny from fishtree (lines 367-374): “Since information on phylogeny was available only for a subset of 4,930 fish species covered in our study (based on Betancur-R et al.⁶⁰), we allocated the remaining species to the phylogenetic tree using an imputation procedure implemented in the R package “fishtree”⁶¹. The empirical tree covered 97% of the families and 80% of the genera included in our species set, indicating that the majority of the missing species were allocated to the correct genus. Our final dataset for the regression included 9,779 species (695 species were excluded because covariates were not available and 951 because they were not included in the “fishtree” database).” Since the imputed phylogeny allocates missing species stochastically, we performed 100 replicates of the phylogenetic regression model per scenario and reported mean and standard deviation across the replicates (see Figure 5). We modified the methods description accordingly (lines 340-389).

Specific comments:

1.3) L90-93: I didn’t understand these two sentences, although I have the feeling that this is related to my previous comment regarding the maximum dispersal scenario (?).

We removed this part as it is not relevant anymore (the dispersal scenario has been adjusted to address comment 1.1).

1.4) L146-148: Perhaps it might be worth developing this point further (briefly).

Thanks for the suggestion. We have completed the sentence as follows (lines 149-152): “However, flow and water temperature seasonality were clearly correlated to the minimum flow and water temperature extremes within the species’ ranges (Pearson’s r of 0.6-0.9), indicating that the effects of changes in the seasonality of flow or temperature were at least partially covered by our predictions.”

1.5) L279 Change ‘modelling’ for ‘projecting’?

This has been amended.

1.6) L292 Wouldn’t ‘at risk’ be a better choice than ‘threatened’?

In this case we prefer the use of ‘threatened’ to be consistent with the terminology used throughout the manuscript.

1.7) L299-311 Both 'sub-watershed' and 'sub-basin' are used in this paragraph. I suggest to use 'sub-basin' throughout the manuscript.

As suggested, we now refer to "sub-basins" consistently throughout the manuscript.

1.8) L319 I suggest to replace the subscript 's' by another letter to avoid confusion with the number of species 'S'

We modified this to 'c' for GCM-RCP scenario combination.

1.9) L348-350 This is a detail but is the information provided in fishbase only binary: 'lake' versus 'rivers'? I thought that it was more similar to what is described for the IUCN datasets

This is correct. In Fishbase, habitat information is provided at different levels of aggregation, whereby lake/river represents the coarsest level. We now specify this in the manuscript (line 359-361): "For the remaining species, we extracted information on habitat from Fishbase, where we used the highest level of aggregation of habitat types to classify species found in lakes as lentic and species found in rivers as lotic."

1.10) L356 I suggest to use the plural 'we ran multiple regressions'

Thanks for the suggestion; this has been amended.

1.11) Figure 4. As I understand it, only pixels flagged as threatened by both flow and temperature thresholds are shown. Perhaps showing the total overlap of the two threats in the bottom panel would be more informative as it is ultimately what has been used in the climate threat analysis? At least the legend should more clearly state what is represented by 'both'

Thanks for raising this issue. We now better specify what "both" means: "**Figure 4.** Potentially affected fraction (PAF) of species due to changes in water temperature (top), flow conditions (center) or both (i.e., fraction of species threatened by water temperature and flow extremes simultaneously; bottom) for the 3.2°C warming scenario." It is correct that our analysis shows the overlap, and we present this in Figure 2, while in Figure 4 we are interested in showing the degree to which the threats are overlapping or not. We mention this at lines 107-109: "Our results further show only limited overlap of threats imposed by amplified flow and water temperature extremes, reflecting the dissimilar spatial distribution of both threats (Figure 4 and Tables S3-S4)"

1.12) Figure 5 – The reference should be to Table S5 not S3. I am also unsure of how the Pearson's r reported in the legend were obtained (same question Figure S7).

Thank you for noticing; we fixed the reference to Table S5. In addition we make more explicit now how the Pearson's r is calculated: "The legend includes the ranges across the different warming levels in Pagel's λ and Pearson's r between the predicted and observed percentage of range threatened."

1.13) Figure S11- (b) should be the percentage instead of ratio?

We corrected this to "percentage."

Reviewer #2

I have gone through the response-to-reviewers letter and the revised manuscript. The authors have incorporated my suggestions and addressed my concerns in their revisions. I think this paper would be an excellent contribution to the literature on the global impact of climate change on freshwater fish.

Thank you for the compliment.

2.1) I have just one more concern/question now that the authors have clarified their phylogenetic modeling approach. My concern is about the response variable in the phylogenetic models. The response variable is a proportion (bounded between 0-1). Proportions are notoriously difficult to model without transformations, and because the authors didn't mention performing any model diagnostics/model assumption checks, I am wondering if the residuals might actually be non-normal, which would violate a key assumption of the model. I suggest the authors make a Q-Q plot of the residuals to check for residual normality along with other standard model diagnostics plots (e.g., residual vs. fitted plots to check for linearity and homogeneity in variance). If model residuals are indeed non-normal (e.g., highly skewed), a log-transform of proportion or using log-response-ratio [i.e., $\log(\text{range size threatened by climate change}/\text{range size not threatened by climate change})$] might help improve/ensure normality of residuals and be more appropriate as a response variable.

Thanks for the suggestion. We have checked the residuals using diagnostic Q-Q plots, as suggested, and have subsequently log-transformed the response variable (area of range threatened/total area, see Equations 6-7), which improved the distribution of the residuals (new Figure S13).

Other than the abovementioned concern, I only have a few more minor suggestions and recommendations to help make the paper clearer for readers.

-Xingli Giam

Specific comments (line numbers refer to the ones on the document where changes were tracked)

2.2) L24. Suggest changing to "In a world that is 3.2°C warmer (if current national targets for reductions in greenhouse gas emissions for 2030 under the Paris Agreement are met), ..." to give more context and background into pledges by governments under the Paris Agreement.

Thanks for the suggestion. To comply with abstract word limit of 150 words, we modified the sentence to: "In a 3.2°C warmer world (no further emission cuts after current governments' pledges for 2030), 36% of the species have over half of their present-day geographic range exposed to climatic extremes beyond current levels."

2.3) L85. Suggest clarifying that 3.2°C warming value is the warming limit by the end of the century with 66% confidence (UN Emissions Gap Report 2019, Pg XIX, last bullet point), i.e., there is a 66% chance that warming will be $\leq 3.2^\circ\text{C}$ by 2100 if current unconditional Nationally Determined Contributions (NDC) are fully implemented. L85-86. Suggest changing to "...set at 3.2°C warming value. This value represents the maximum warming limit predicted to occur by the end of the century (with 66% probability) if all current greenhouse gas emissions reductions targets (unconditional Nationally Determined Contributions; NDC) for 2030 are met (cite UN Emissions Gap Report 2019).

Following this suggestion, we have clarified the 3.2°C level (lines 48-53): “For comparison purposes, we include two additional scenarios: a “current pledges” scenario set at 3.2°C warming and a “no-policy” scenario (no mitigation) set at 4.5°C warming (all temperatures relative to pre-industrial)^{12,19}. The 3.2°C warming scenario represents the maximum warming predicted to occur by the end of the century (with 66% probability) if all current greenhouse gas emissions reductions targets (unconditional Nationally Determined Contributions) for 2030 are met and no further cuts are performed.”

2.4) L133-134. “Under the maximal dispersal scenario...”. This section lacks a comparison with the “no dispersal” scenario. Do the authors mean that the maximal and no dispersal scenarios show similar spatial patterns but with slightly lower threat levels distributed over larger areas? If so, then change to “Under the maximal dispersal scenario, threatened areas are similar to those under the no dispersal scenario but with lower threat levels distributed over larger areas than in the no dispersal scenario.”

Thanks for the suggestion. We have modified the sentence accordingly (lines 91-93): “Under the maximal dispersal scenario, locations of threatened areas are similar to those under the no dispersal scenario but with lower threat levels than in the no dispersal scenario (Figure 2 and 3).”

2.5) L149-156. The authors presented results in areas affected by changes in low flow. How about changes in high flow?

We have added the following sentence (lines 105-107): “In contrast, areas affected by changes in high flow are confined to a few downstream segments of the main stems of large rivers (Figure S3 and raster layers provided as SI).”

2.6) L157. Does “amplified flow” here mean both low and high flow? It might be confusing because on L144, the authors mentioned “amplified low or high flow conditions”.

We changed “high and low” to “extreme” to avoid confusion and be consistent throughout the section.

2.7) L179. I looked at Table S5-II and interestingly, the threat to climate change was positively associated with the (ordinal levels) of threat status. Coefficients were always LC < NT < VU < EN < 0 (CR ref. level) with the exception of 1.5°C warming scenario. Even then LC and NT coefficients were very similar. I wonder if this is worth mentioning in the results and the discussion.

Thank you for the suggestion. As recommended, we now mention this in the results (lines 119-122): “We also noticed lower threat levels for species currently belonging to a low IUCN threat category (e.g., “near threatened” or “least concern”; Table S5). We found similarly low threat levels for species that are “data deficient” within the IUCN Red List (43% of the 9,779 species analyzed).” And in the discussion (lines 177-179): “Species already listed as “endangered” or “critically endangered” on the IUCN Red List of threatened species might be particularly affected by future warming, as these species were characterized by the highest future climate threat levels.”

2.9) Table S5-II. In the caption, I suggest including information with which factor level is the reference level by indicating for example, A=Equatorial (reference level); CR=Critically endangered (reference level), etc.

Thank you for the suggestion; we have added this.

2.10) L202-203. Suggest changing “amplified high water temperatures” to “the increase in maximum water temperatures”.

It has been amended.

2.11) L204. Suggest replacing “This reflects that” with “This is because”.

Adjusted as suggested.

2.12) L226. Suggest insert “explicitly” between “did not” and “consider”.

It has been added.

2.13) L230. The author can consider inserting at the end of the current sentence “, indicating that the effects of seasonal flow or temperature changes were likely at least partially incorporated in our predictions.”

Thanks for the suggestion; we have incorporated this notion (see also our reply to comment 1.4).

2.14) L253. Correct typo error: replace “treat” with “threat”.

Amended.

2.15) L547. From what I understand, the authors ran only a phylogenetic regression model, right? If that is the case, replace “multiple linear regression analysis” with “phylogenetic regression analysis”. Combine the first and second sentences on this page: “We performed a phylogenetic regression analysis to relate...to a number of potentially relevant species characteristics while accounting for the non-independence of observations due to phylogenetic relatedness among species.”

The term “multiple regression model” indicates that the regression model included multiple covariates. In any case, we also ran more than one regression model, eight in total (four warming targets x two dispersal scenarios), so the term “multiple” might be confusing here and we removed it as suggested. We also substituted linear with phylogenetic and combined the first two sentences as suggested.

2.16) L548. The response variable is a proportion value (bounded between 0-1). This is potentially an issue because the model residuals are often far from normally distributed, which is an important assumption of the phylogenetic regression model. The authors did not mention checking of model assumptions and or performing standard model diagnostics. I suggest the authors make a Q-Q plot of the model residuals to check for normality of residuals and plot residuals against fitted values to check for linearity and homogeneity in variance. From experience, modeling proportions might be really tricky. If the residuals are not normally distributed, log-transforming the proportions might work in some cases but not all. If that doesn’t work, using the log-response-ratio [i.e., log (range size threatened by climate change/range size not threatened by climate change)] might work better.

Thank you for this suggestion. We have log-transformed our response variable, which improved the distribution of the residuals. See further our reply to 2.1.

2.17) L573. Suggest replacing “multiple regression” with “individual phylogenetic regression models”.

See reply to 2.15. We have modified it to phylogenetic regression models.

Reviewers' Comments:

Reviewer #1:

Remarks to the Author:

The authors have taken care to address all my remaining concerns. I comment on the authors persistence in addressing all the points raised during the review process - the changes have strengthen the relevance of the work and this study will undoubtedly be a significant contribution to the field.

For the sake of transparency I signed my report.

Lise Comte